# SLAKE: Softmax-Approximated Training-Free Linear Attention with KV-Cache Eviction for Long-Sequence LLMs

## Abstract

Recent advances in transformer-based large language models (LLMs) have enabled inference over contexts as long as 128K tokens. However, the quadratic computational and memory costs of full self-attention remain a fundamental bottleneck at such scales. Prior efforts to mitigate this challenge largely fall into two camps: (i) structural approximations (*e.g.*, linear attention) that reduce asymptotic complexity but typically require costly retraining, and (ii) KV-cache optimizations (*e.g.*, eviction or merging) that are training-free yet inevitably discard information. We introduce **S**oftmax-Approximated Training-Free **L**inear **A**ttention with **K**V-Cache **E**viction (*SLAKE*), a novel framework that unifies the complementary advantages of these two paradigms. At its core, *SLAKE* employs Partially Taylor-Approximated Attention (PTAA), which leverages a first-order Taylor expansion to selectively linearize the Softmax attention kernel. This design enables tokens deemed low-importance via eviction scoring to be processed efficiently with linear attention, while preserving exact Softmax computation for high-salience tokens. To further improve cache efficiency, we propose Value-Aware Budget Scoring (VABS), a new allocation strategy that incorporates value contributions and overcomes key limitations of previous eviction heuristics. Extensive experiments on LLaMA-3 8B demonstrate that *SLAKE* delivers up to $10\times$ inference speedup and 30.8% peak-memory reduction on 128K-token sequences, while keeping accuracy loss below 4%. To our knowledge, *SLAKE* is the first training-free approach to jointly integrate linear attention with KV-cache eviction, establishing a new state of the art among long-context, training-free methods.

## 1 Introduction

Recently, the transformer architecture has become the foundational backbone of natural language processing (NLP) and is widely adopted in large language models (LLMs) Radford et al. (2019); Touvron et al. (2023). In line with this trend, state-of-the-art LLMs such as Llama-3 Grattafiori et al. (2024) and GPT-4 Achiam et al. (2023) support inference with input lengths exceeding 128K tokens to enable long-sequence tasks, including document summarization Zhang et al. (2024), multi-turn dialogues Chiang et al. (2023), information retrieval Liu et al. (2023), and question answering Kamalloo et al. (2023). Consequently, the development of models and computational platforms capable of handling such long sequences has emerged as a critical research direction to enhance both the practicality and performance of AI systems. However, the self-attention mechanism, a core component of the transformer architecture, suffers from quadratic computational complexity due to its inherent dependence on query, key, and value interactions, where the computation grows as $O(N^2)$ with respect to the input sequence length $N$ Beltagy et al. (2020). As a result, LLMs encounter severe hardware bottlenecks, with computational cost and memory usage increasing exponentially as the input length increases Wang et al. (2020).

To alleviate these challenges, two major lines of research have been explored: (i) linear-complexity attention and (ii) KV-cache compression. First, Performer Choromanski et al. (2020) and Linearized LLM You et al. (2024) introduced linear attention mechanisms with $O(N)$ computational complexity, thereby improving efficiency. These methods also leverage the characteristics of linear attention to maintain KV-caches with fixed sizes, effectively mitigating memory bottlenecks. In a different

direction, Mamba Gu & Dao (2023) proposed a state space model (SSM)-based architecture as an alternative to self-attention, offering an additional solution to the quadratic complexity problem. However, this approach fundamentally alters the attention mechanism, preventing the reuse of pretrained weights and inevitably requiring retraining (***Limit.*** ①). Consequently, such methods require substantial computational resources and time, eventually falling short of fully supporting long-sequence tasks. In contrast, KV-cache compression methods Zhang et al. (2023); Yang et al. (2024); Qin et al. (2025); Nawrot et al. (2024) evaluate the importance of tokens in real time, removing or merging less important tokens to reduce the number of tokens involved in attention. This approach can alleviate both computational complexity and memory bottlenecks; however, the removal or corruption of certain tokens inevitably leads to performance degradation (***Limit.*** ②). DMC Nawrot et al. (2024) combines token eviction and merging and performs retraining to compensate for lost information, but the finetuning process is computationally intensive and demands substantial GPU resources, making it challenging to apply across diverse tasks. Furthermore, the dynamic budget allocation scores used in existing KV-cache eviction techniques do not account for the influence of the value component in self-attention. As a result, the eviction criteria fail to fully reflect the quality of attention approximation (***Limit.*** ③).

While linear attention and KV-cache eviction approaches have limitations in terms of training overhead and accuracy degradation due to token information loss, they remain effective means for mitigating computational and memory bottlenecks. Linear attention replaces Softmax with low-dimensional kernel features, preserving global content but reducing token-level fidelity, which yields measurable performance drops Choromanski et al. (2020). By contrast, KV-cache eviction retains exact Softmax on a selected subset of tokens, yet irreversibly discards information from evicted entries, harming accuracy Zhang et al. (2023). A natural strategy is therefore to combine their complementary strengths: keep high-importance tokens for precise Softmax attention via eviction, while approximating low-importance tokens with linear attention to reduce overhead. However, two obstacles still impede this integration: (1) Training overhead—linear attention introduces structural changes that typically require retraining; and (2) Kernel incompatibility—most linear kernels deviate substantially from Softmax, making direct combination with Softmax-based eviction prone to accuracy loss. As a result, simultaneously leveraging both paradigms without degrading Softmax behavior—or incurring substantial additional training and hardware cost—remains challenging.

To leverage the complementary characteristics of linear attention and KV-cache eviction, this work proposes the Softmax-Approximated Training-Free Linear Attention with KV-Cache Eviction (*SLAKE*) framework. In *SLAKE*, a Partially Taylor Approximated Attention (PTAA) mechanism processes a limited set of important tokens using precise attention, while the remaining tokens are handled via Taylor approximation-based linear attention, resulting in a more accurate approximation of Softmax attention compared to conventional linear attention methods. Furthermore, by employing the Taylor approximation, *SLAKE* preserves the validity of pretrained weights, addressing the key limitation of prior linear attention approaches that require additional training. In addition, this study introduces Value Aware Budget Scoring (VABS), which enhances existing dynamic budget allocation schemes by considering the influence of the value matrix on attention outcomes. VABS enables more precise cache budget allocation, improving the accuracy of attention approximation. By combining these two techniques, *SLAKE* effectively maintains the performance of existing LLMs without additional training while significantly reducing computational cost and memory usage, representing the first instance of integrating linear attention with KV-cache eviction. Experimental results demonstrate that *SLAKE* achieves high accuracy with less than a 4% drop compared to the LongBench baseline on Llama2-7B Touvron et al. (2023), Llama3.1-8B Grattafiori et al. (2024), and Mistral-7B-v0.3 Jiang et al. (2023). Moreover, on GPUs with 128K tokens, *SLAKE* attains a $10\times$ inference speedup and a 30.83% reduction in peak memory usage relative to the baseline. The main contributions of this work are summarized as follows:

- Solution for ***Limit.*** ①: We propose a train-free first-order Taylor approximation to transform standard self-attention into a linear attention format. This approach effectively reproduces the functionality of self-attention without requiring additional retraining, alleviating the training burden inherent in previous linear attention models.
- Solution for ***Limit.*** ②: We introduce PTAA, an attention structure that integrates linear attention with KV-cache eviction, for the first time. PTAA maintains the computational and memory efficiency of conventional KV-cache compression while improving performance on long-sequence tasks.

- Solution for **Limit.** ③: We identify a key limitation of existing KV-cache eviction methods, as they fail to consider the impact of the value component in self-attention. To address this, we propose a novel budget scoring mechanism, VABS, which dynamically allocates cache budgets while taking the value influence into account.

## 2 BACKGROUND

### 2.1 SELF-ATTENTION OPERATION AND KV CACHING IN AUTOREGRESSIVE DECODING

The core operation of the transformer is self-attention, which captures contextual information by computing relationships among all tokens within the input sequence Vaswani et al. (2017). Given an input sequence of length $N$ and embedding dimension $d$, self-attention is defined as follows:

$$\text{Attention}(Q, K, V) = \text{Softmax}\left(\frac{QK^\top}{\sqrt{d}}\right)V, \tag{1}$$

Here, $Q, K, V \in \mathbb{R}^{N \times d}$ denote the query, key, and value matrices, respectively, where $Q = [q_1, \ldots, q_N]^\top$, $K = [k_1, \ldots, k_N]^\top$, and $V = [v_1, \ldots, v_N]^\top$ consist of token embedding rows $q, k, v$. Self-attention computes the attention score for each token pair $(i, j)$, with $i, j \in 1, \ldots, N$, by taking the dot product of the $i$-th token's query $q_i$ and the $j$-th token's key $k_j$. Then, self-attention normalizes the scores using Softmax and then takes a weighted sum of the values $v_j$ to produce new token representations. To enhance the representational capacity of a single self-attention layer, modern LLMs employ multi-head self-attention (MHSA). MHSA splits the input token embeddings into $h$ separate heads, performs self-attention independently within each head, and then concatenates the resulting representations. MHSA is formally defined as follows:

$$\text{MHSA}(Q, K, V) = \text{Concat}(\text{head}_1, \ldots, \text{head}_h)W^O, \tag{2}$$

The attention output of each head, $\text{head}_m$ with $m \in 1, \ldots, h$, is computed as follows:

$$\text{head}_m = \text{Attention}(QW_m^Q, KW_m^K, VW_m^V) \tag{3}$$

Here, $W_m^Q, W_m^K, W_m^V \in \mathbb{R}^{d \times d_h}$ and $W^O \in \mathbb{R}^{d \times d}$ are learnable linear projection weights, where $d_h = d/h$ is typically used to evenly split the dimension for each head. This allows the model to learn diverse contextual information across different representation subspaces, resulting in richer semantic representations Voita et al. (2019). However, both self-attention and MHSA require computing similarities for all token pairs $(i, j)$, with $i, j \in 1, \ldots, N$, leading to a computational complexity of $O(N^2 d)$ and a memory complexity of $O(N^2)$. As the input sequence length $N$ increases in long-sequence tasks, both computation and memory usage grow rapidly, becoming a major bottleneck for LLMs. Meanwhile, LLMs typically operate in an autoregressive decoding setting for long-sequence tasks, where the output at timestep $t - 1$ serves as the input for timestep $t$. In this context, the query $q_t$ at timestep $t$ must interact with all past keys and values, which can be expressed as follows:

$$h_t = \text{Softmax}\left(\frac{q_t K_{1:t}^\top}{\sqrt{d_h}}\right)V_{1:t}. \tag{4}$$

In other words, at each timestep, the query must be computed against all previous keys and values, causing the recomputation cost of $K_{1:t-1}$ and $V_{1:t-1}$ to grow exponentially as the sequence length increases. To mitigate this, KV caching was introduced Dai et al. (2019), which stores keys and values from previous timesteps in a cache. Using KV caching, when a new token query $q_t$ is generated, only $q_t$ is computed, and the corresponding $k_t$ and $v_t$ are concatenated to the existing cache as:

$$K_{1:t} = [K_{1:t-1}; k_t], \quad V_{1:t} = [V_{1:t-1}; v_t]. \tag{5}$$

This approach allows only the new key ($k_t$) and value ($v_t$) to be appended at each timestep, thereby reducing redundant computations. However, the size of the KV-cache still grows linearly with the sequence length, $O(Nd)$, and in modern models processing inputs of 128K tokens or more, storing the cache itself becomes a significant memory bottleneck.

### 2.2 PREVIOUS KV-CACHE COMPRESSION METHODS

Prior KV-cache compression methods mitigate both the computational burden of self-attention and the memory overhead of the cache by limiting participation to a subset of tokens, with method-specific criteria determining which tokens are retained. Streaming LLM Xiao et al. (2023) exploits

the observation that early tokens in attention tend to carry relatively higher importance, maintaining only the initial and most recent tokens. This approach demonstrates that conventional LLMs can support sequence lengths approaching infinity. However, its performance degrades significantly as the cache budget decreases. The Heavy-Hitter Oracle (H2O) Zhang et al. (2023) goes beyond retaining only recent tokens by evaluating token importance based on self-attention scores. It selects 'heavy-hitter tokens' while removing the remaining ones, thereby reducing unnecessary memory usage. Nevertheless, H2O does not consider cache budgets across Transformer blocks, which can result in severe performance degradation. PyramidKV Yang et al. (2024) proposes a method to minimize performance loss by gradually increasing the KV-cache budget across transformer blocks. However, its fixed budget allocation cannot account for variations in input data, leading to significant performance fluctuations depending on the input. CAKE Qin et al. (2025) not only dynamically allocates per-layer cache budgets based on the entropy and variance of the self-attention map according to input tokens, but also introduces a cascading KV-cache eviction strategy that maintains the overall KV-cache budget consistently from the prefill stage, thereby minimizing performance degradation due to token removal. Nonetheless, accuracy loss persists as information from evicted tokens is inevitably discarded. DMC Nawrot et al. (2024) proposes a dynamic memory compression technique that decides whether to evict or merge tokens based on their characteristics, and employs fine-tuning to minimize performance degradation. However, the hardware and temporal overhead imposed by fine-tuning cannot be ignored.

### 2.3 PREVIOUS LINEAR-COMPLEXITY ATTENTION METHODS

Existing linear-complexity attention methods aim to alleviate the computational burden while preserving the ability of self-attention to capture interactions between all tokens. Linformer Wang et al. (2020) addresses the $O(N^2)$ computational bottleneck of transformers by leveraging a low-rank approximation of the attention matrix. Based on the observation that the majority of information in the attention matrix is concentrated in a few dominant singular values, it employs a linear projection layer to project the entire sequence into a fixed, shorter sequence. This linear projection layer is applied to the key and value matrices, significantly reducing the attention computation cost and lowering the computational complexity to $O(Nk)$. However, because Linformer employs a fixed linear projection layer after training, its performance can degrade substantially under fluctuations in input activation. Moreover, it requires training from scratch and does not support modern LLM architectures. Longformer Beltagy et al. (2020) introduces a sparse attention mechanism to mitigate the $O(N^2)$ complexity when processing long sequences. In this sparse attention structure, each token attends only to a fixed-size window of neighboring tokens ($w$), reducing the complexity to $O(N \times w)$. Additionally, selected important tokens receive global attention across the entire sequence to integrate contextual information. Similar to Linformer, Longformer requires model training and does not support contemporary LLM architectures. Performer Choromanski et al. (2020) alleviates the $O(N^2)$ complexity of standard self-attention by approximating the Softmax function with a kernel function $\phi(\cdot)$, which can be separately applied to queries and keys. This allows the self-attention computation to be reformulated as a linear-complexity attention mechanism as follows:

$$\text{Attention}(Q, K, V) \approx \phi(Q) \left( \phi(K)^\top V \right) \tag{6}$$

Through the reformulation of computations as in Eq. 6, linear attention reduces the matrix multiplication complexity between queries, keys, and values to $O(N)$. However, Performer does not support modern model architectures and exhibits severe performance degradation due to rank collapse of the attention map, which arises from approximating long-sequence attention with a simple kernel function Yu et al. (2022); Han et al. (2023). Linearized LLM You et al. (2024) applies linear attention to the LLama2-7B and 13B architectures Touvron et al. (2023) and, separately from linear attention, employs depthwise convolution Howard et al. (2017) on the value matrix to restore the lost rank of the attention map. As the first study to apply linear attention to relatively modern LLMs, such as LLama2-7B and 13B, Linearized LLM also supports speculative decoding, enabling multiple tokens to be predicted simultaneously using smaller models. Nevertheless, this approach requires training from scratch and still suffers from non-negligible performance degradation despite the rank restoration. Meanwhile, the Mamba model Gu & Dao (2023), based on SSM architectures, can effectively capture long-range dependencies without using attention or MLP layers. Mamba introduces a selective state space mechanism that dynamically adjusts the parameters of the state space model according to the input, achieving high performance with linear time complexity even for long-sequence tasks.

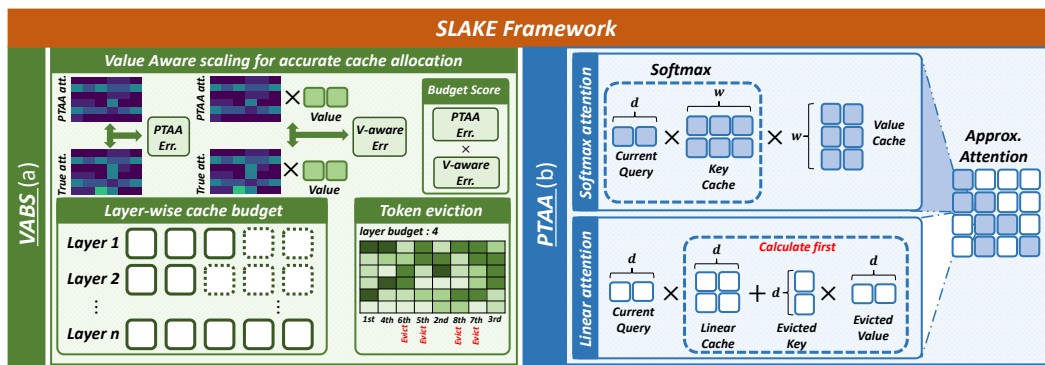

Figure 1: Overall SLAKE architecture. (a) VABS allocates a layerwise cache budget for each transformer layer by taking into account both the linear approximation error and the value data. (b) PTAA combines linear attention with Softmax attention to effectively emulate standard self-attention.

## 3 PROPOSED METHOD: *SLAKE*

### 3.1 OVERVIEW OF *SLAKE*

Figure 1 illustrates the overall process of *SLAKE*. In the prefill stage, *SLAKE* first performs VABS, which considers both the linear approximation error of the attention map and the contribution of value data (for **Limit.** ③). Based on these scores, layerwise cache budgets are allocated, and token eviction is applied to adjust the KV-cache size to the target budget. In the generation stage, *SLAKE* evaluates newly generated tokens together with existing ones and incrementally stacks them in the linear cache. PTAA is then applied: evicted tokens are approximated using linear attention, while preserved tokens are processed with softmax attention. Through this process, *SLAKE* adapts to the characteristics of input tokens without requiring additional training (for **Limit.** ①) and effectively approximates self-attention at each generation step (for **Limit.** ②).

### 3.2 TRAIN-FREE LINEAR ATTENTION VIA TAYLOR APPROXIMATION

Unlike previous linear attention methods, *SLAKE* leverages a first-order Taylor approximation to linearly approximate self-attention without any additional training. This approach allows pretrained weights to mitigate information loss caused by approximating the Softmax attention operation Jin et al. (2025). In other words, the weights learned based on the complex nonlinearity of the Softmax operation remain effective even under the linearized computation provided by the first-order Taylor approximation. This enables *SLAKE* to perform linear attention using existing pretrained weights without requiring any retraining (**Limit.** ① solved). Let the query of the $i$th token be denoted as $q_i$, the keys and values including all past tokens as $K$ and $V$, each element of the keys and values as $k$ and $v$, and the attention head dimension as $d_h$. Then, self-attention can be expressed as follows:

$$\text{Att}(q_i, K, V) = \sum_{j=1}^{n} \frac{\exp\left(\frac{q_i^\top k_j}{\sqrt{d_h}} - \max_{1 \leq l \leq n}(x_{i,l})\right)}{\sum_{l=1}^{n} \exp\left(\frac{q_i^\top k_l}{\sqrt{d_h}} - \max_{1 \leq l \leq n}(x_{i,l})\right)} v_j \qquad (7)$$

Next, we apply a first-order Taylor approximation to the exponential function in Softmax attention. To this end, let the product between $q_i$ and $k$ be denoted as $x_{i,j}$, the mean of $x_{i,j}$ as $x_{i,\text{mean}}$, and the maximum of $x_{i,j}$ as $x_{i,\text{max}}$, defined as follows:

$$x_{i,j} = \frac{q_i^\top k_j}{\sqrt{d_h}}. \qquad (8)$$

$$x_{i,\text{mean}} = \frac{1}{n}\sum_{l=1}^{n} x_{i,l} = \frac{1}{n}\sum_{l=1}^{n} \frac{q_i^\top k_l}{\sqrt{d_h}} = \frac{1}{n}q_i^\top \sum_{l=1}^{n} \frac{k_l}{\sqrt{d_h}} \qquad (9)$$

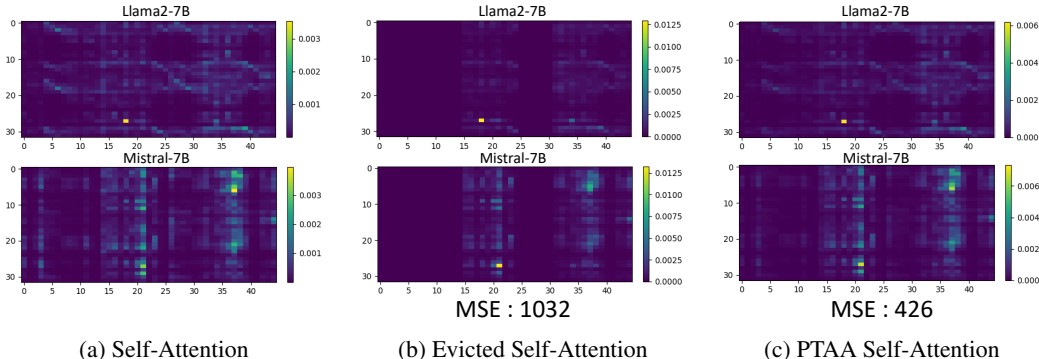

(a) Self-Attention      (b) Evicted Self-Attention      (c) PTAA Self-Attention

Figure 2: Comparison of Self-Attention, Evicted Self-Attention, and PTAA Self-Attention

$$x_{i,\max} = \max_{1\leq l\leq n}(x_{i,l}) = \max_{1\leq l\leq n}\Big(\frac{q_i^\top k_l}{\sqrt{d_h}}\Big) \tag{10}$$

This allows the exponential function to be expressed as a first-order polynomial as follows:

$$\exp(x_{i,j}) \approx \exp_t(x_{i,j}) = \exp(x_{i,\mathrm{mean}} - x_{i,\max})\Big(1 + x_{i,j} - x_{i,\mathrm{mean}}\Big) \tag{11}$$

Applying the Taylor approximation of the exponential function to the entire Softmax operation, the computation of self-attention can then be expressed as follows:

$$\mathrm{Att}(q_i, K, V) \approx \sum_{j=1}^{n} \frac{\exp(x_{i,\mathrm{mean}} - x_{i,\max})\Big(v_j + q_i^\top \frac{k_j}{\sqrt{d_h}}v_j - x_{i,\mathrm{mean}}v_j\Big)}{n(x_{i,\mathrm{mean}} - x_{i,\max})} . \tag{12}$$

In this case, the dependency between $q_i^\top$ and $k_j$ in Eq. 12 is removed, allowing the computation between $k_j$ and $v_j$ to be performed in advance, resulting in a linear attention format. This enables a train-free transformation to linear attention. A more detailed proof can be found in Appendix A.

## 3.3 Merging Linear Attention with KV-cache Eviction

We propose the approximation method (*i.e.*, PTAA) to simultaneously leverage the global token processing capability of the train-free linear attention described in the previous section and the focused token processing capability of KV-cache eviction. PTAA determines, via KV-cache eviction scoring, whether each token should be processed by Taylor-approximated linear attention or by Softmax-based attention. For this purpose, we adopt the CAKE score Qin et al. (2025) as the eviction score. Let $\hat{K}, \hat{V}$ denote the KV-cache for linear attention after eviction at timestep $t$, $K, V$ denote the KV-cache for Softmax attention, and $\hat{x}_i = \frac{q_i^\top \hat{k}_j}{\sqrt{d_h}}$. Then, the PTAA computation can be expressed as follows (see Appendix B for a detailed proof):

$$E_i = \exp(x_i - x_{i,\max}), \quad \hat{E}_i = \exp(\hat{x}_{i,\mathrm{mean}} - x_{i,\max}) \tag{13}$$

$$\mathrm{PTAA}(q_i, K, V) = \sum_{j=1}^{n} \frac{E_i + \hat{E}_i\Big(\sum_{j=1}^{n}\hat{v}_j + \frac{q_i^\top}{\sqrt{d_h}}(\hat{k}_n v_n + \sum_{j=1}^{n-1}\hat{k}_j\hat{v}_j) - \frac{1}{n}q_i^\top\sum_{l=1}^{n}\frac{\hat{k}_l}{\sqrt{d_h}}\sum_{j=1}^{n}\hat{v}_j\Big)}{n(E_i + \hat{E}_i)} \tag{14}$$

In this case, both terms are independent of the timestep: the linear attention term has a complexity of $O(d^2)$, and the eviction-based Softmax term has a complexity of $O(wd)$ with respect to the cache budget $w$. Consequently, the overall computation is simplified from the original $O(Nd)$ to $O((d + w)d)$ and no longer depends on the sequence length $N$. Moreover, our proposed method enables the utilization of information from past tokens that would otherwise be discarded in conventional KV-cache eviction methods, thereby significantly improving approximation accuracy (***Limit.*** ② solved).

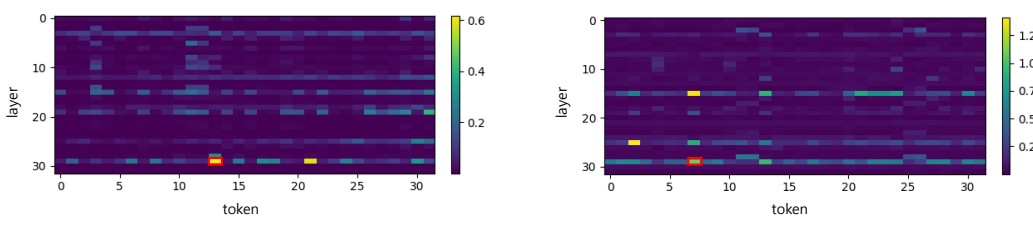

(a) Token-wise Error After PTAA      (b) Token-wise Error After Value Multiplication

Figure 3: Comparison of the PTAA approximation error and the error after multiplication with the value matrix in the Llama2-7B model.

Figure 2 visually and quantitatively demonstrates how PTAA improves approximation accuracy compared to standard eviction. Specifically, Figure 2(a) shows the attention map without eviction, (b) shows the result after eviction, and (c) shows the attention map with PTAA applied in *SLAKE*. Notably, in Figure 2(b), information from the left and central tokens is completely removed (zeroed out), whereas in Figure 2(c), this information is clearly restored. As a result of this restoration, the mean squared error of PTAA is reduced by more than half compared to the token eviction method 1032. The actual operation of PTAA during text generation can be found in Appendix C.

### 3.4 VALUE AWARE BUDGET SCORING

*SLAKE* analyzes the limitations of existing dynamic cache allocation methods and proposes a new scoring approach for dynamic cache budget allocation. Conventional scoring methods typically leverage the entropy or variance of the query-key attention map to reflect token distribution tendencies across layers. However, since self-attention computations inherently depend on interactions among queries, keys, and values, these scoring methods fail to fully account for the influence of the value matrix. Empirical results illustrate this limitation: as shown in Figure 3(a), the PTAA approximation error in the 30th layer is largest at the 13th token. After multiplication with the value matrix, however, the largest error shifts to the 7th token, as seen in Figure 3(b). This demonstrates that approximation errors in self-attention can be significantly amplified by the value matrix.

Therefore, in this work, we propose VABS, a novel scoring metric that accounts not only for approximation errors induced by PTAA but also for errors amplified by the value matrix. To use the PTAA approximation error as the base score, VABS computes the total variation between the PTAA output and the true attention output as follows:

$$\text{TV}_{hiq} = \frac{1}{2}\sum_j |\tilde{Att}_{t,hij} - Att_{a,hij}| \tag{15}$$

where $\tilde{Att}$ denotes the attention output with PTAA applied, and $Att$ denotes the Softmax attention output. The error term ValueAware, which accounts for the influence of the value matrix $V$ on the PTAA error, is then computed as follows:

$$\text{ValueAware}_{hi} = \left(\sum_j \frac{\|V_{hij}\|}{\sum_{j'}\|V_{hij'}\|} \cdot |\tilde{Att}_{t,hij} - Att_{a,hij}|\right)^{\gamma} \tag{16}$$

Here, a scaling term $\gamma$ is set to 0.1 to control the influence of the error term. Finally, the entropy term (Entr) reflecting the concentration of the attention map's probability distribution is computed as:

$$\text{Entr}_{hi} = -\sum_j Att_{t,hij} \log Att_{t,hij} \tag{17}$$

Finally, after applying scaling factors $\alpha$ and $\beta$, the value-aware preference score is computed by multiplying the mean of the product of the total variation term and the value-aware term with the entropy term, as follows. Detailed values of the scaling factors for each model can be found in Appendix D.

$$\text{PrefScore} = \left(\mathbb{E}_{,h,i}[\text{TV}_{hi} \cdot \text{ValueAware}_{hi}]\right)^{\alpha} \cdot \left(\mathbb{E}_{h,i}[\text{Entr}_{hi}]\right)^{\beta}. \tag{18}$$

Table 1: Comparison of LongBench performance between SLAKE and existing methods across various models

| Model | Method | Cachesize | Single-Document QA | | | Multi-Document QA | | | Summarization | | | Few-shot Learning | | | Synthetic | | Code | | Avg. |
|---|---|---|---|---|---|---|---|---|---|---|---|---|---|---|---|---|---|---|---|
| | | | NrtvQA | Qasper | MF-en | HotpotQA | 2WikiMQA | Musique | GovRepor | QMSum | MultiNews | TREC | TriviaQA | SAMSum | PCount | PR-en | Lcc | RB-P | |
| Falcon Mamba-7B | - | - | 7.04 | 31.54 | 30.11 | 25.78 | 27.66 | 10.26 | 23.16 | 21.22 | 25.76 | 70.0 | 78.74 | 34.5 | 0.00 | 4.50 | 48.55 | 39.24 | 29.88 |
| Llama2-7B chat | Full KV | - | 18.74 | 21.41 | 37.57 | 27.78 | 32.03 | 7.53 | 26.85 | 20.97 | 26.41 | 64.00 | 83.59 | 41.37 | 2.85 | 7.00 | 60.48 | 54.39 | 33.31 |
| | H2O | 128 | 14.23 | 14.78 | 26.51 | 20.05 | 28.73 | 5.00 | 16.00 | 20.00 | 20.64 | 38.00 | 72.47 | 37.10 | 2.30 | 3.50 | 49.69 | 47.37 | 26.05 |
| | PyramidKV | | 13.45 | 16.61 | 32.52 | 22.76 | 28.59 | 7.56 | 18.87 | 19.96 | 20.05 | 43.50 | 79.90 | 36.74 | 2.29 | 8.00 | 51.19 | 47.56 | 28.09 |
| | CAKE | | 14.27 | 16.50 | 31.87 | 24.27 | 27.08 | 7.60 | 19.11 | 20.64 | 20.63 | 47.00 | 80.53 | 37.83 | 3.25 | 8.00 | 54.11 | 50.25 | 28.93 |
| | SLAKE(ours) | | 13.94 | 15.60 | 31.24 | 25.84 | 30.77 | 7.73 | 19.64 | 20.35 | 20.61 | 54.50 | 80.60 | 37.85 | 3.24 | 8.50 | 55.29 | 50.52 | 29.76 |
| | H2O | 256 | 15.27 | 15.31 | 27.24 | 21.02 | 27.34 | 6.31 | 19.75 | 20.45 | 22.23 | 45.51 | 79.64 | 37.93 | 2.93 | 4.00 | 52.45 | 51.23 | 28.04 |
| | PyramidKV | | 15.13 | 15.86 | 33.93 | 25.58 | 29.51 | 9.23 | 20.35 | 21.22 | 22.01 | 58.00 | 82.39 | 38.47 | 2.15 | 7.50 | 55.81 | 49.53 | 30.42 |
| | CAKE | | 15.38 | 15.62 | 35.55 | 26.92 | 30.04 | 9.18 | 20.39 | 20.92 | 22.34 | 58.00 | 81.97 | 38.91 | 2.72 | 6.50 | 58.03 | 52.48 | 30.93 |
| | SLAKE(ours) | | 16.36 | 15.69 | 35.79 | 27.77 | 30.77 | 8.92 | 20.40 | 20.71 | 22.61 | 61.50 | 82.08 | 39.04 | 2.46 | 7.50 | 58.87 | 52.36 | 31.43 |
| Llama3.1-8B chat | Full KV | - | 23.67 | 12.97 | 27.48 | 17.00 | 16.44 | 11.98 | 34.26 | 23.41 | 26.80 | 72.50 | 91.55 | 44.09 | 6.70 | 96.78 | 65.17 | 58.85 | 39.35 |
| | H2O | 128 | 16.24 | 4.70 | 21.70 | 12.40 | 11.30 | 6.83 | 19.34 | 20.34 | 19.30 | 35.00 | 85.40 | 41.90 | 5.40 | 89.04 | 58.23 | 51.34 | 31.15 |
| | PyramidKV | | 16.43 | 5.20 | 20.30 | 13.12 | 11.90 | 6.54 | 20.74 | 21.13 | 20.60 | 46.00 | 88.10 | 42.03 | 6.50 | 90.11 | 59.24 | 51.34 | 32.46 |
| | CAKE | | 17.04 | 6.34 | 21.60 | 14.43 | 13.08 | 8.85 | 20.56 | 21.34 | 20.6 | 48.00 | 87.96 | 42.03 | 7.08 | 90.81 | 60.60 | 52.01 | 33.27 |
| | SLAKE(ours) | | 18.51 | 7.26 | 21.89 | 14.16 | 13.86 | 9.13 | 21.84 | 21.50 | 20.96 | 52.00 | 90.02 | 41.25 | 6.83 | 94.36 | 60.57 | 52.79 | 34.18 |
| | H2O | 256 | 18.34 | 5.41 | 23.13 | 12.90 | 13.24 | 8.11 | 20.14 | 21.43 | 21.11 | 52.00 | 90.34 | 42.10 | 6.14 | 90.45 | 62.13 | 52.34 | 33.71 |
| | PyramidKV | | 21.34 | 7.14 | 21.69 | 14.79 | 15.70 | 7.25 | 23.24 | 22.45 | 22.14 | 59.00 | 90.01 | 41.10 | 7.01 | 90.13 | 63.24 | 54.91 | 35.07 |
| | CAKE | | 20.07 | 8.50 | 23.04 | 15.36 | 15.40 | 8.89 | 23.11 | 22.71 | 23.29 | 59.50 | 91.11 | 41.47 | 7.07 | 95.00 | 62.87 | 55.60 | 35.81 |
| | SLAKE(ours) | | 21.41 | 9.21 | 23.67 | 14.82 | 14.07 | 9.00 | 22.97 | 22.55 | 22.89 | 58.00 | 91.43 | 42.42 | 7.71 | 96.16 | 63.38 | 55.19 | 35.93 |
| Mistral-7B-v0.3 | Full KV | - | 28.55 | 38.30 | 50.02 | 51.94 | 36.14 | 26.00 | 33.86 | 25.50 | 26.63 | 76.00 | 88.64 | 47.22 | 4.50 | 97.00 | 61.37 | 62.88 | 47.16 |
| | H2O | 128 | 22.34 | 23.13 | 42.11 | 43.99 | 28.14 | 19.43 | 20.34 | 22.98 | 19.34 | 43.50 | 86.24 | 42.72 | 2.00 | 92.13 | 53.90 | 52.13 | 38.40 |
| | PyramidKV | | 23.91 | 24.89 | 40.07 | 41.76 | 30.43 | 19.14 | 22.04 | 21.98 | 20.09 | 44.00 | 88.64 | 43.10 | 3.00 | 92.45 | 55.80 | 53.23 | 39.03 |
| | CAKE | | 24.75 | 26.31 | 43.11 | 45.93 | 31.11 | 20.46 | 22.89 | 22.77 | 20.99 | 45.50 | 89.66 | 43.14 | 3.50 | 92.00 | 56.23 | 55.43 | 40.24 |
| | SLAKE(ours) | | 25.44 | 28.89 | 45.34 | 45.49 | 31.45 | 22.69 | 22.34 | 22.87 | 20.75 | 52.00 | 88.52 | 43.47 | 4.00 | 93.50 | 56.61 | 56.21 | 41.22 |
| | H2O | 256 | 26.03 | 27.01 | 44.23 | 43.88 | 32.55 | 21.55 | 24.16 | 24.10 | 21.46 | 46.12 | 88.36 | 45.22 | 5.00 | 94.57 | 56.02 | 55.35 | 40.98 |
| | PyramidKV | | 25.28 | 26.07 | 43.01 | 46.93 | 31.08 | 22.08 | 23.28 | 25.92 | 23.03 | 46.44 | 89.58 | 45.66 | 4.50 | 95.07 | 58.74 | 55.07 | 41.49 |
| | CAKE | | 24.90 | 29.46 | 47.37 | 47.87 | 33.40 | 20.78 | 24.67 | 23.35 | 22.67 | 57.00 | 89.44 | 44.20 | 4.00 | 94.00 | 58.94 | 59.71 | 42.61 |
| | SLAKE(ours) | | 25.45 | 31.06 | 45.79 | 47.16 | 34.66 | 22.99 | 24.23 | 23.59 | 22.70 | 63.00 | 89.23 | 44.90 | 4.50 | 96.50 | 59.55 | 59.90 | 43.45 |

Through this approach, VABS effectively captures how errors in the attention map are amplified by the value matrix, thereby overcoming the limitations of conventional scoring methods based solely on the query-key attention map. Consequently, cache budgets can be allocated in a way that better reflects the actual impact on model performance (**Limit.** ③ solved). Using VABS, *SLAKE* can allocate cache budgets that reflect both the characteristics of PTAA approximations and the influence of the value matrix, minimizing PTAA-induced errors.

# 4 EXPERIMENTAL RESULTS

## 4.1 EXPERIMENTAL SETTINGS

To evaluate the performance of *SLAKE*, we conducted experiments on a range of large language models, including Llama3.1-8B Grattafiori et al. (2024), Llama2-7B Touvron et al. (2023), and Mistral-7B-v0.3 Jiang et al. (2023). The comparison set includes linear-complexity attention models such as Falcon Mamba Zuo et al. (2024) as well as KV-cache optimization methods, including H2O Zhang et al. (2023), Pyramid KV Yang et al. (2024), and CAKE Qin et al. (2025). Evaluation was performed using the long-sequence task benchmark LongBench Bai et al. (2023). LongBench comprises 16 datasets across six categories, including single- and multi-document question answering, summarization, few-shot learning, synthetic tasks, and code completion, focusing specifically on assessing long-context understanding. Additionally, to analyze the hardware efficiency of *SLAKE*, we measured latency and peak memory usage on an NVIDIA H100 GPU as a function of sequence length. The comparative experiments were conducted with average KV-cache budgets of 128 and 256, and to ensure a fair comparison, all methods were constrained to use the same total cache size.

## 4.2 LONGBENCH RESULTS

To evaluate the performance of *SLAKE* against existing KV-cache optimization methods and linear attention-based approaches, we conducted experiments on all 16 LongBench datasets. Table 1 reports the task-wise performance under cache budgets of 128 and 256. The results show that *SLAKE* not only significantly outperforms existing linear-complexity attention methods but also achieves superior performance compared to other KV-cache optimization methods using the same cache budget. In particular, on the Llama2-7B model, *SLAKE* consistently surpasses all conventional KV-cache methods under both the 128 and 256 budget settings. Similarly, on the Llama3.1-8B and Mistral-7B-v0.3 models, *SLAKE* achieves higher average accuracy across all KV-cache optimization methods.

Table 2: Comparison of Long-Bench average scores for Llama2-7B with PTAA, dynamic cache, and VABS

| Eviction | PTAA | VABS | score |
|:---:|:---:|:---:|:---:|
| ✓ | ✗ | ✗ | 28.81 |
| ✓ | ✓ | ✗ | 29.47 |
| ✓ | ✓ | ✓ | 29.76 |

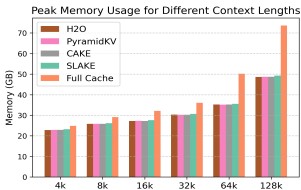

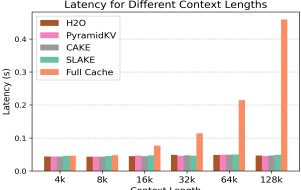

(c): Peak Memory Usage for Different Context Lengths

(d): Latency for Different Context Lengths

Figure 4: Peak memory usage and decoding latency on H100 80GB.

### 4.3 ABLATION STUDIES

Next, we describe the LongBench experimental results conducted to validate the effectiveness and compatibility of PTAA and VABS proposed in *SLAKE*. Table 2 compares the performance on Llama2-7B under three settings: applying only the dynamic cache budget allocation-based eviction from the original CAKE, adding PTAA, and applying dynamic budget allocation via VABS. When only eviction was applied, the average score remained at 28.81. Adding PTAA increased the score to 29.27, demonstrating that PTAA effectively restores token information that would otherwise be removed during the eviction process. Finally, applying dynamic cache budget allocation based on VABS yielded the highest score of 29.76, indicating that VABS appropriately accounts for the impact of value on errors arising from self-attention approximations. These results confirm the effectiveness of both PTAA and VABS individually. Moreover, when the two algorithms are integrated, they complement each other to achieve the highest accuracy, effectively optimizing errors that may arise during cache budget allocation and the approximation process.

### 4.4 PEAK MEMORY AND THROUGHPUT EVALUATION

We evaluated the hardware efficiency of *SLAKE* by measuring peak memory usage and decoding latency during LLM inference on a FlashAttention-2 Dao (2023) enabled Llama-3.1-8B-Instruct Grattafiori et al. (2024) model. The comparisons included full cache, H2O Zhang et al. (2023), PyramidKV Yang et al. (2024), and CAKE Qin et al. (2025), with all methods maintaining a uniform cache budget of 1024 to ensure a fair comparison. As shown in Figure 4(a), *SLAKE* achieves memory savings comparable to existing KV-cache optimization methods, reducing peak memory usage by approximately 30.83% at a 128K context length compared to full cache. Figure 4(b) illustrates that *SLAKE* also achieves latency on par with other KV-cache optimization techniques. Notably, in the 128K context length setting, *SLAKE* reduces decoding latency by more than $10 \times$ relative to full cache. These results demonstrate that *SLAKE* effectively alleviates the computational and memory bottlenecks inherent in full cache settings Dao et al. (2022).

## 5 CONCLUSION

In this work, we propose a novel framework, *SLAKE*, which combines linear attention with KV-cache eviction to simultaneously improve efficiency and accuracy in long-context LLM inference. *SLAKE* employs the PTAA attention approximation method, which processes important tokens using exact attention while applying a Taylor-based approximation only to less important tokens, achieving Softmax-level performance without additional training. Furthermore, by introducing the VABS scoring method that accounts for the influence of the value matrix, *SLAKE* allocates cache resources more precisely across layers, minimizing errors arising from the approximation. Through extensive experiments, we demonstrate that *SLAKE* consistently outperforms existing linear attention and KV-cache compression methods in long-context models. This design provides a practical approach for long-context inference, significantly reducing computation and memory usage while maintaining full-attention-level performance.

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

APPENDIX

## A PROOF OF TRAIN-FREE LINEAR ATTENTION VIA TAYLOR APPROXIMATION

In this section, we provide a detailed explanation of how self-attention can be transformed into a linear attention format using Taylor approximation. By applying Taylor expansion to the self-attention operation described in Section 3.2, we can rewrite it as follows. Specifically, the exponential function can be approximated in the form of a first-order polynomial. Under this approximation, the operation of the softmax function $\tilde{Att}$ can be expressed as shown below, where we denote $\sum_{l=1}^n k_l = k_{\text{sum}}$ and $\sum_{l=1}^n v_l = v_{\text{sum}}$.

$$
\begin{aligned}
\tilde{Att}(q_i, K, V) &= \sum_{j=1}^n \frac{\exp\left(\frac{1}{n}\sum_{l=1}^n \frac{q_i^\top k_l}{\sqrt{d_h}} - x_{i,max}\right)\left(1 + \frac{q_i^\top k_j}{\sqrt{d_h}} - \frac{1}{n}\sum_{l=1}^n \frac{q_i^\top k_l}{\sqrt{d_h}}\right)}{\sum_{m=1}^n \exp\left(\frac{1}{n}\sum_{l=1}^n \frac{q_i^\top k_l}{\sqrt{d_h}} - x_{i,max}\right)\left(1 + \frac{q_i^\top k_m}{\sqrt{d_h}} - \frac{1}{n}\sum_{l=1}^n \frac{q_i^\top k_l}{\sqrt{d_h}}\right)} v_j \\
&= \sum_{j=1}^n \frac{\exp\left(\frac{1}{n}\frac{q_i^\top}{\sqrt{d_h}}\sum_{l=1}^n k_l - x_{i,max}\right)\left(v_j + \frac{q_i^\top}{\sqrt{d_h}}k_j v_j - \frac{1}{n}q_i^\top \sum_{l=1}^n \frac{k_l}{\sqrt{d_h}}v_j\right)}{\sum_{m=1}^n \exp\left(\frac{1}{n}\frac{q_i^\top}{\sqrt{d_h}}\sum_{l=1}^n k_l - x_{i,max}\right)} \\
&= \frac{\exp\left(\frac{1}{n}\frac{q_i^\top}{\sqrt{d_h}}k_{sum} - x_{i,max}\right)\left(v_{sum} + \frac{q_i^\top}{\sqrt{d_h}}(k_n v_n) - \frac{1}{n}\frac{q_i^\top}{\sqrt{d_h}}k_{sum}v_{sum}\right)}{\sum_{m=1}^n \exp\left(\frac{1}{n}\frac{q_i^\top}{\sqrt{d_h}}k_{sum} - x_{i,max}\right)}
\end{aligned}
$$

(19)

Through this process, the computational dependency between $q$ and $k$ in self-attention is removed, enabling its transformation into a linear attention format. Strictly speaking, $x_{i,\max}$ is defined as $\max_{1 \le l \le n}\left(\frac{q_i^\top k_l}{\sqrt{d_h}}\right)$, so a dependency on this term still remains. However, since PTAA employs the maximum value from the exact attention term as the reference maximum, the dependency is fully eliminated in PTAA.

## B PROOF OF PTAA

In this section, we explain how the PTAA operation can be linearized. Specifically, we first evaluate the attention map $A^t$ within the given cache budget using the previous KV-cache $K^{t-1}, V^{t-1}$ together with the newly added key and value at timestep $t$, namely $k^t$ and $v^t$. Based on this evaluation, we identify the tokens to be evicted, denoted as $\hat{K}$ and $\hat{V}$, and update the KV-cache to $K^t, V^t$. These updated components are then utilized in PTAA, which can be expressed as follows:

$$
\text{Evict}\left([K^{t-1}, k^t], [V^{t-1}, v^t], A^t\right) = (\hat{K}, \hat{V}, K^t, V^t)
$$

(20)

Subsequently, depending on the classification, $\hat{K}$ and $\hat{V}$ are processed with linear attention, while $K$ and $V$ are handled with softmax attention in parallel, which can be expressed as follows:

$$
= \frac{\exp(x_i - x_{i,\max}) + \exp\left(\frac{1}{n}\frac{q_i^\top}{\sqrt{d_h}}\hat{k}_{sum} - x_{i,\max}\right)\left(\hat{v}_{sum} + \frac{q_i^\top}{\sqrt{d_h}}(\hat{k}_n v_n + \sum_{j=1}^{n-1}\hat{k}_j \hat{v}_j) - \frac{1}{n}\frac{q_i^\top}{\sqrt{d_h}}\hat{k}_{sum}\hat{v}_{sum}\right)}{\exp(x_i - x_{i,\max})n + \sum_{m=1}^n \exp\left(\frac{1}{n}\frac{q_i^\top}{\sqrt{d_h}}\hat{k}_{sum} - x_{i,max}\right)}
$$

(21)

At this stage, since $x_{i,\max}$ is independent of the tokens used in linear attention, the linear attention operation is able to maintain linear complexity.

---

**Algorithm 1:** PTAA-Based Autoregressive Decoding Process

---

**Input:** Layer index $l$, query $Q \in \mathbb{R}^{h \times 1 \times d_h}$, key cache $K \in \mathbb{R}^{h \times len_{cache} \times d_h}$, value cache
$V \in \mathbb{R}^{h \times len_{cache} \times d_h}$, linear cache $L \in \mathbb{R}^{h \times d_h \times d_h}$, linear K sum $k_{\text{sum}} \in \mathbb{R}^{h \times 1 \times d_h}$,
linear V sum $v_{\text{sum}} \in \mathbb{R}^{h \times 1 \times d_h}$, eviction score $E_s$, head dimension $d_h$, cache size
$len_{cache}$, linear cache size $\ell_a$, KV group num $g$, KV group size $h$

**Output:** Attention output AttnOut

// evict caches
$K, k_e \leftarrow \text{evict}(E_s, K)$;
$V, v_e \leftarrow \text{evict}(E_s, V)$;
// update linear cache
$k_{\text{sum}} \leftarrow k_{\text{sum}} + k_e$;
$v_{\text{sum}} \leftarrow v_{\text{sum}} + v_e$;
$L \leftarrow L + k_e^\top v_e$;
// expand caches
$K \leftarrow \text{repeat\_kv}(K, g)$;
$V \leftarrow \text{repeat\_kv}(V, g)$;
linear_cache $\leftarrow \text{repeat\_kv}(\text{linear\_cache}, g)$;
$k_{\text{sum}} \leftarrow \text{repeat\_kv}(k_{\text{sum}}, g)$;
$v_{\text{sum}} \leftarrow \text{repeat\_kv}(v_{\text{sum}}, g)$;
// Compute standard attention weights
$\text{Att} \leftarrow \dfrac{QK^\top}{\sqrt{d_h}}$;
$a_{\max} \leftarrow \max(\text{Att}, \text{axis} = -1)$;
$\text{Att} \leftarrow \exp(\text{Att} - a_{\max})$;
$s_{\text{att}} \leftarrow \sum \text{Att}$;
// Compute linear approximation term
$\text{Att}_{\text{lin}} \leftarrow \dfrac{QL}{\sqrt{d_h}}$;
$\mu \leftarrow \dfrac{Qk_{\text{sum}}^\top}{\ell_a \sqrt{d_h}}$;
$\lambda \leftarrow \exp(\mu - a_{\max})$;
$\text{Att}_{\text{lin}} \leftarrow (\text{Att}_{\text{lin}} + (1 - \mu)v_{\text{sum}}) \cdot \lambda$;
// Combine outputs
$\text{Att} \leftarrow \text{Att}V$;
$\text{AttnOut} \leftarrow \dfrac{\text{Att} + \text{Att}_{\text{lin}}}{s_{\text{att}} + \lambda \ell_a}$;
**return** AttnOut;

---

## C PTAA ALGORITHM

Algorithm 1 illustrates how *SLAKE*, based on PTAA, operates during the autoregressive decoding process. Here, $Q$ denotes the attention query, while $K$ and $V$ represent the cached keys and values. $L$ refers to the Linear Cache, which is the cumulative sum of the matrix products of keys and values that have been evicted in the past. Similarly, $k_{\text{sum}}$ and $v_{\text{sum}}$ denote the cumulative sums of the evicted keys and values, respectively. In the PTAA process, the eviction score $E_s$ is first used to remove $k_e$ and $v_e$ from the existing KV-cache $K$ and $V$. The removed $k_e$ and $v_e$ are then added to the cumulative sums $k_{\text{sum}}$ and $v_{\text{sum}}$, and their product $k_e^\top v_e$ is added to $L$. Since data of the same size is added to the Linear Cache at each step, the overall cache size remains constant. Afterward, each cache is expanded according to the head group size. Attention is then performed over $Q$ and $K$, followed by linear attention based on the Taylor approximation. Finally, the PTAA process is completed by combining the result of softmax attention, $Att$, with that of linear attention, $Att_{\text{lin}}$. Because the key–value product is added to the Linear Cache at every decoding step, the cache size remains unchanged. Moreover, the size of the linear cache itself is $d_h^2$, which is negligible compared to the original KV-cache.

Table 3: Comparison of $\alpha$ and $\beta$ by the model and cache budget

| model | Llama2-7B | | Llama3.1-8B | | Mistral-7B-v0.3 | |
|---|---|---|---|---|---|---|
| cache budget | 128 | 256 | 128 | 256 | 128 | 256 |
| $\alpha$ | 0.5 | 0.5 | 0.4 | 0.6 | 0.6 | 0.4 |
| $\beta$ | 0.4 | 0.4 | 0.2 | 0.5 | 0.5 | 0.2 |

Table 4: Needle in the haystack multi-reasoning results

| model | method | cache size | Multi-Reasoning | | |
|---|---|---|---|---|---|
| | | | EN | ZH | Overall |
| Llama3-8B-Chat | Full KV | – | 84.20 | 69.50 | 76.85 |
| | SLAKE (ours) | 128 | 46.69 | 13.72 | 30.21 |
| | | 256 | 49.47 | 27.08 | 38.28 |
| Mistral-7B-Instruct-v0.3 | Full KV | – | 79.13 | 65.33 | 72.23 |
| | SLAKE (ours) | 128 | 48.87 | 17.98 | 33.43 |
| | | 256 | 52.11 | 28.00 | 40.06 |

## D    VABS SCALING TERM IN VARIOUS LLM MODELS

In this section, we present the values of the scaling terms $\alpha$ and $\beta$ used in VABS for each model, as summarized in Table 3.

## E    NEEDLE IN THE HAYSTACK BENCHMARK RESULTS

In this section, we evaluate the performance of *SLAKE* on the NeedleBench Li et al. (2024) task to verify its capability in retrieval and multi-step reasoning under complex contexts. Specifically, we conduct the Multi-Needle Reasoning benchmark with an 8K context length, comparing *SLAKE* against the Full KV baseline under cache budgets of 128 and 256. As shown in Table 4, *SLAKE* demonstrates stable performance even with limited cache resources, and at a cache budget of 256, it recovers more than half of the Full KV performance. These results confirm that *SLAKE* can effectively handle complex reasoning tasks under constrained memory conditions.

