# OpenReview forum: "SLAKE: Softmax-Approximated Training-Free Linear Attention with KV-Cache Eviction for Long-Sequence LLMs"
_ICLR.cc/2026/Conference — Submitted to ICLR 2026_

### Official Review · Reviewer_4zgC · 2025-10-28

**Soundness:** 3
**Presentation:** 2
**Contribution:** 3
**Rating:** 6
**Confidence:** 2

**Summary:**

This work proposes a framework that improves the efficiency of long-context inference in large language models without retraining. It introduces Partially Taylor-Approximated Attention (PTAA), which applies a first-order Taylor expansion to partially linearize the Softmax attention kernel so that low-importance tokens can be processed with linear attention while high-importance tokens retain exact Softmax computation. To complement this, it presents Value-Aware Budget Scoring (VABS), a dynamic cache-budget allocation method that accounts for both the approximation error of PTAA and the influence of the value matrix when selecting tokens for eviction. Together, PTAA and VABS combine the advantages of linear attention and KV-cache compression in a training-free manner. Experiments on LLaMA-2-7B, LLaMA-3.1-8B, and Mistral-7B-v0.3 using the LongBench benchmark show that SLAKE achieves up to a 10× inference speedup and 30.8% peak-memory reduction for 128K-token sequences while maintaining less than 4% accuracy loss, establishing a new state-of-the-art among training-free long-context methods.

**Strengths:**

1. The paper introduces a novel combination of linear attention and KV-cache eviction into a single training-free framework. Its Partially Taylor-Approximated Attention and Value-Aware Budget Scoring components offer new ways to linearize Softmax and allocate cache budgets by explicitly modeling value contributions.
2. The paper is well-structured, and the main ideas and methodology are easy to follow.

**Weaknesses:**

1. The ablation study is not sufficiently comprehensive to fully support the individual contributions of PTAA and VABS. The current results show average LongBench scores under three configurations (eviction only, +PTAA, +VABS on LLaMA2-7B with a 128-token cache budget).
2. The experimental design lacks sensitivity analyses for the approximation hyperparameters (e.g., Taylor truncation order, scaling factors in VABS). Reporting how performance and stability vary with these parameters would clarify the robustness of the method.
3. Several experimental details are missing, including decoding hyperparameters such as temperature, maximum generation length, and top-p values. Reporting these settings would improve the reproducibility and interpretability of the experimental results.

**Questions:**

See the Weaknesses section.

---

> ### Author Response · Authors · 2025-11-21
>
> We thank the reviewer for their valuable comments.
> ### **W1) Ablation Study on PTAA and VABS Contributions**
> > - As the reviewer pointed out, the original ablation study alone did not sufficiently demonstrate that PTAA and VABS each independently contribute to performance improvement. To address this, we conducted the same ablation experiments not only on Llama-2-7B chat but also on the Mistral-7B-v0.3 model. Additionally, we included a setting where VABS was applied to the existing eviction method without PTAA, allowing us to clearly analyze the contributions of the two techniques. In this case, the approximate self-attention required for VABS was computed using the Eviction-based attention rather than PTAA, effectively separating the effects of the two components.
> > **Table A** Ablation Study Results of PTAA and VABS Across Models
> | Model             | Eviction | PTAA | VABS | Score |
> |------------------|----------|------|------|-------|
> | Llama-2-7B chat  | X        | X    | X    | 33.31 |
> | Llama-2-7B chat  | O        | X    | X    | 28.81 |
> | Llama-2-7B chat  | O        | O    | X    | 29.47 |
> | Llama-2-7B chat  | O        | X    | O    | 29.16 |
> | Llama-2-7B chat  | O        | O    | O    | 29.76 |
> | Mistral-7B-v0.3  | X        | X    | X    | 47.16 |
> | Mistral-7B-v0.3  | O        | X    | X    | 40.24 |
> | Mistral-7B-v0.3  | O        | O    | X    | 40.65 |
> | Mistral-7B-v0.3  | O        | X    | O    | 40.87 |
> | Mistral-7B-v0.3  | O        | O    | O    | 41.22 |
>
> > - As shown in the results below, both Llama2-7B chat and Mistral-7B-v0.3 demonstrate that PTAA and VABS individually provide meaningful performance improvements and exhibit consistent trends. This clearly indicates that the two components of SLAKE independently and complementarily enhance attention quality.

---

> ### Author Response · Authors · 2025-11-21
>
> ### **W2) Sensitivity Analysis of Approximation Hyperparameters**
> > - As requested by the reviewer, we conducted a hyperparameter sensitivity analysis for both the Taylor truncation order and VABS.
> > - First, to implement linear attention using Taylor approximation, it is essential to apply a first-order Taylor approximation. While using a second-order or higher Taylor approximation may improve approximation accuracy, it significantly reduces inference efficiency to a level comparable with FullKV. In other words, for linear Taylor approximation, the truncation order must always be 1. The performance using the second-order Taylor approximation is shown below.
> > - We also conducted experiments on VABS hyperparameters α and β. The tuning load for VABS hyperparameters (α, β, γ) is not significant. In all our experiments, we restricted the search range for α and β to [0.1, 0.7] and used a fixed value of 0.5 for γ. To determine suitable α and β values, we performed a calibration using 50 samples from each of the five major LongBench tasks. Each calibration step takes approximately 7 minutes on an H100 GPU, and after 36 calibration steps, the final α and β values are determined, requiring a total of about 4 hours. Additionally, we analyzed the performance variation of α and β on Llama2-7B chat. As shown in the [Link1], the grid search results indicate that the highest average score is achieved around [0.5, 0.4], and performance remains consistently high within this range.
> > - To further minimize the cost of VABS hyperparameter tuning, we implemented an α–β prediction algorithm leveraging the model configuration. Specifically, we used the model’s embedding dimension, GQA group size, and average cache budget as inputs to a second-order linear regression to predict α and β. Using this method, we determined the hyperparameters [0.2, 0.1] for the larger Qwen2.5-32B model, which reliably enabled VABS functionality and achieved superior performance compared to CAKE.
> > **Table B** Qwen2.5-32B LongBench Performance with Automatically Estimated VABS Hyperparameters
> | Model                  | Mthd        | Cache | NQA | Qspr | MF | HQA | 2WMQA | Msq | GvRpr | QMSm | MltNs | TRC | TrvQ | SMSm | PCnt | PR | Lcc | RB-P | Avg. |
> |------------------------|---------------|-----------|--------|--------|-------|----------|----------|---------|----------|-------|-----------|------|----------|--------|--------|-------|-----|------|------|
> | Qwen2.5-32B instruct   | Full KV       | -         | 29.25  | 45.24  | 51.90 | 63.31    | 61.50    | 38.15   | 30.33    | 23.38 | 23.01     | 73.50 | **87.86** | 45.36  | 12.00  | 100.00 | 51.30 | 38.17 | 48.39  |
> | Qwen2.5-32B instruct   | H2O           | 128       | 24.03  | 24.48  | 31.10 | 52.53    | 52.38    | 30.84   | 17.00    | 19.67 | **16.27** | 43.00 | 83.64  | 41.41  | 12.22  | 94.58  | 45.85 | 34.35 | 38.96  |
> | Qwen2.5-32B instruct   | PyramidKV     | 128       | 18.31  | 24.62  | 36.85 | 55.30    | 55.40    | 32.38   | 15.33    | 18.02 | 13.15     | 46.50 | **84.50** | 39.62  | 11.50  | 88.83  | 44.05 | 31.76 | 38.51  |
> | Qwen2.5-32B instruct   | CAKE          | 128       | 24.03  | 29.35  | **41.64** | 57.42    | 57.37    | 31.85   | 18.54    | **20.42** | 16.26     | 50.00 | 82.90  | 40.63  | **12.50** | 97.42  | **46.59** | 33.86 | 41.30  |
> | Qwen2.5-32B instruct   | SLAKE (ours)  | 128       | **26.84** | **35.89** | 41.54 | **57.81** | **58.58** | **33.41** | **19.74** | 18.84 | 15.79     | **63.50** | 72.37  | **42.53** | **12.50** | **99.08** | 46.37 | **35.54** | **42.52** |
>
> ### **References**
> [Link1] https://github.com/wasdfgsd/ICLR2026-15640/blob/main/Llama2-7B_VABS.png

---

> ### Author Response · Authors · 2025-11-21
>
> ### **W3) Decoding Hyperparameters and Experimental Settings**
> > - We thank the reviewer for the comment.
> > - The default sampling parameters used in all experiments were set as follows:
> >> - Temperature: 1
> >> - Top-p: None
>
> In addition, the maximum generation length was set differently for each task, as specified below:
>
> | Task                        | Max Generation Length |
> |------------------------------|---------------------|
> | narrativeqa                  | 128                 |
> | qasper                       | 128                 |
> | multifieldqa_en              | 64                  |
> | multifieldqa_zh              | 64                  |
> | hotpotqa                     | 32                  |
> | 2wikimqa                     | 32                  |
> | musique                      | 32                  |
> | dureader                     | 128                 |
> | gov_report                   | 512                 |
> | qmsum                        | 512                 |
> | multi_news                   | 512                 |
> | vcsum                        | 512                 |
> | trec                         | 64                  |
> | triviaqa                     | 32                  |
> | samsum                       | 128                 |
> | lsht                         | 64                  |
> | passage_count                | 32                  |
> | passage_retrieval_en         | 32                  |
> | passage_retrieval_zh         | 32                  |
> | lcc                          | 64                  |
> | repobench-p                  | 64                  |

---

> ### Comment · Area_Chair_9KYM · 2025-11-23
>
> Dear reviewer,
>
> Thanks for your time and effort in reviewing ICLR2026 submissions. The authors have submitted their responses to your review. Please take the time to read and raise your further comments, and discuss with the authors.
>
> Best regards,
>
> AC

---

> > ### Author Response · Authors · 2025-12-03
> >
> > ### **Summary of Rebuttal to Reviewer 4zgC**
> >
> > Dear Area Chair and Reviewers,
> >
> > We appreciate Reviewer 4zgC’s positive evaluation and constructive feedback regarding the isolation of component contributions and hyperparameter robustness. In our response, we have provided expanded ablation studies and sensitivity analyses to solidify the validity of our proposed methods.
> >
> > **1. Comprehensive Ablation Study (Isolating PTAA and VABS)**
> > Addressing the concern that the individual contributions of PTAA and VABS were not sufficiently distinct, we conducted additional experiments:
> > * **Expanded Scope:** We performed ablation studies on Mistral-7B-v0.3 in addition to Llama-2-7B.
> > * **Component Isolation:** We introduced a new experimental setting where VABS is applied to standard eviction without PTAA. The results (Table A) clearly demonstrate that VABS and PTAA provide independent and additive performance gains, confirming their orthogonal contributions to the SLAKE framework.
> >
> > **2. Hyperparameter Sensitivity and Automation**
> > * **Taylor Expansion Order:** We clarified that a first-order Taylor expansion is mathematically necessary to maintain the linear complexity benefits of our approach. Higher-order approximations would negate the efficiency gains, reverting the cost to quadratic levels.
> > * **VABS Stability & Automation:** We demonstrated that VABS performance is stable within a broad range ($\alpha, \beta \in [0.1, 0.7]$). Furthermore, to resolve concerns about tuning overhead, we implemented an automated regression algorithm that predicts optimal hyperparameters based on model configuration. This automated setup was successfully validated on Qwen2.5-32B, achieving SOTA performance without manual tuning.
> >
> > **3. Reproducibility and Experimental Details**
> >
> > We provided the full set of decoding hyperparameters (Temperature = 1, Top-p = None) and task-specific generation lengths requested by the reviewer to ensure full reproducibility of our results.
> >
> > We believe these additional data points resolve the reviewer's uncertainties regarding the robustness and distinct value of SLAKE's components.

---

### Official Review · Reviewer_MgG5 · 2025-10-31

**Soundness:** 3
**Presentation:** 3
**Contribution:** 3
**Rating:** 6
**Confidence:** 4

**Summary:**

This paper proposes SLAKE, a training-free framework to address the quadratic complexity bottleneck in LLM long-sequence inference. SLAKE is the first training-free approach to jointly integrate linear attention with KV-cache eviction. Its core PTAA mechanism uses exact Softmax computation for high-salience tokens while processing low-importance tokens with a Taylor-approximated linear attention. Furthermore, it introduces VABS, a novel cache allocation strategy that incorporates the influence of the Value matrix to overcome limitations of previous eviction heuristics. Experiments show SLAKE outperforms existing training-free methods on LongBench, achieving up to a 10x inference speedup and a 30.8% peak-memory reduction on 128K-token sequences .

**Strengths:**

1. The paper introduces SLAKE, the first framework to be training-free while unifying two distinct optimization paradigms: linear attention and KV-cache eviction. Its core mechanism, PTAA, innovatively retains information from evicted tokens via Taylor approximation instead of discarding it, which directly addresses the information loss problem of standard eviction methods.
2. SLAKE achieves state-of-the-art results, consistently outperforming other training-free eviction methods like H2O and CAKE on the comprehensive LongBench benchmark. This accuracy gain is driven in part by VABS, a more insightful scoring metric that corrects a key flaw in prior methods by accounting for the Value matrix's influence.

**Weaknesses:**

1. Ambiguous Computational Cost: The paper is unclear about the prefill stage computational cost. Both VABS (requiring "true attention") and PTAA (requiring $x_{i,max}$) could hide an $O(N^2)$ step, which would weaken the efficiency claims.Weak Retrieval
2. Performance: On NeedleBench (Table 4), SLAKE's performance drops significantly compared to Full KV, suggesting the approximation struggles with high-fidelity information retrieval tasks.
3. Hyperparameters: VABS introduces hyperparameters $\alpha$ and $\beta$ (Table 3), and the paper does not discuss their sensitivity or the cost of tuning them.
4. Limited Model Scale Validation: The paper's experiments are confined to 7B and 8B models. While sufficient to compare against other training-free eviction methods, cited related work (like Linearized LLM) has explored 13B models. This lack of validation on larger-scale models leaves the method's scalability in question.

**Questions:**

1. VABS Cost: How is the VABS score (Eq. 15), which requires the "true attention output," computed during the prefill stage without incurring an $O(N^2)$ cost?

2. PTAA Cost: How is the $x_{i,max}$ normalization term for the Taylor approximation calculated? Is it from all $N$ tokens (implying $O(N^2)$) or only the $w$ kept tokens (which would be an inaccurate normalizer for the evicted tokens)?

3. Prefill vs. Decoding: Does the 10x speedup refer only to the $O(N)$ decoding phase, or is it an end-to-end time including prefill?

4. VABS Tuning: How sensitive is VABS performance to the $\alpha$ and $\beta$ hyperparameters? What is the cost of tuning these for a new model?

---

> ### Author Response · Authors · 2025-11-21
>
> We thank the reviewer for their valuable comments.
> ### **W1, Q1) $O(N^2)$ Cost of PTAA and VABS in Prefill Stage**
> > - Reviewer’s concern regarding hidden $O(N^2)$ computational costs does not apply to the SLAKE architecture. In the prefill stage, both VABS and PTAA do not perform full attention over all $N$ tokens. Instead, they operate only on the nearest 32 tokens ($N \times 32$) used for eviction scoring. As a result, the computational complexity of both components is limited to $O(N \cdot 32)$, and the typical $O(N^2)$ overhead of full attention never arises. Moreover, all values required for these computations are directly reused from the existing eviction token scoring process, which already computes the attention map ($N \times 32$), making the operations highly efficient.
> > - In terms of actual cost, for Llama2-7B-chat with a context length of 4,096, PTAA introduces 137.88 MFLOPs and VABS adds 113.25 MFLOPs during the prefill stage—both negligible fractions of the total inference workload. During decoding, the cost of PTAA further drops to 1.97 MFLOPs, which is insignificant compared to the total computation of a single Llama2-7B self-attention layer (68,850.13 MFLOPs).
>
> ### **Q2) Cost of PTAA in Decoding Stage**
> > - PTAA computes the normalization term $\(x_{i,\mathrm{mean}}\)$ using the accumulated sum of the keys and queries from the evicted tokens, as shown below:
>
>  $x_{i,\mathrm{mean}} = \frac{1}{n} \frac{q_i^\top}{\sqrt{d_h}} k_{\mathrm{sum}} .$
>
> > - Specifically, when the embedding dimension is $\(d\)$, the accumulated key sum $\(k_{\mathrm{sum}}\)$ has shape $\((\text{batch}, \text{head}, 1, d)\)$ at each generation step, and each query vector $\(q_i\)$ also has shape $\((\text{batch}, \text{head}, 1, d)\)$. As a result, computing $\(x_{i,\mathrm{mean}}\)$ involves $\(O(d^2)\)$ operations.
> > - Thus, PTAA performs accurate normalization using evicted tokens while avoiding the $\(O(N^2)\)$ complexity of full attention, enabling efficient computation.
>
> ### **W2) Performance of SLAKE on Needlebench**
> > - Thank you for your valuable insight. To better evaluate the practical utility of SLAKE, we conducted additional comparative experiments with H2O, CAKE, and SLAKE under identical cache budgets (128/256) using the Llama3-8B-Chat and Mistral-7B-Instruct-v0.3 models.
> > - As shown in Table D, SLAKE consistently outperforms existing methods at the same cache size.
>
> > **Table D. NeedleBench – Multi-Reasoning Performance under Cache Constraints**
> | Model                      | Method        | Cache Size | EN     | ZH     | Overall |
> |---------------------------|--------------|------------|--------|--------|---------|
> | Llama3-8B-Chat         | Full KV       | -          | 84.05  | 68.88  | 76.46   |
> | Llama3-8B-Chat            | H2O           | 128        | 54.52  | 7.82   | 31.17   |
> | Llama3-8B-Chat            | CAKE          | 128        | 64.23  | 11.73  | 37.98   |
> | Llama3-8B-Chat            | **SLAKE** | 128    | 65.63  | 12.49  | 39.06   |
> | Llama3-8B-Chat            | H2O           | 256        | 64.93  | 18.38  | 41.66   |
> | Llama3-8B-Chat            | CAKE          | 256        | 69.74  | 23.31  | 46.53   |
> | Llama3-8B-Chat            | **SLAKE** | 256    | 70.52  | 23.78  | 47.15   |
> | Mistral-7B-Instruct-v0.3 | Full KV    | -          | 84.05  | 68.88  | 76.47   |
> | Mistral-7B-Instruct-v0.3  | H2O           | 128        | 44.47  | 11.92  | 28.20   |
> | Mistral-7B-Instruct-v0.3  | CAKE          | 128        | 48.37  | 16.62  | 32.50   |
> | Mistral-7B-Instruct-v0.3  | **SLAKE** | 128    | 49.59  | 17.72  | 33.66   |
> | Mistral-7B-Instruct-v0.3  | H2O           | 256        | 51.84  | 21.52  | 36.68   |
> | Mistral-7B-Instruct-v0.3  | CAKE          | 256        | 54.15  | 23.52  | 38.84   |
> | Mistral-7B-Instruct-v0.3  | **SLAKE** | 256    | 54.89  | 23.34  | 39.12   |

---

> ### Author Response · Authors · 2025-11-21
>
> ### **W3, Q4) VABS Hyperparameter Analysis and Tuning Cost**
> > - The hyperparameters α, β, and γ of VABS do not impose a significant tuning burden. In all of our experiments, α and β were restricted to the range [0.1, 0.7], and γ was fixed at 0.5. To determine appropriate values for α and β, we performed a lightweight calibration using 50 samples from each of the five major LongBench tasks. On an H100 GPU, each calibration step took approximately 7 minutes, and the final values were obtained within about 4 hours after 36 calibration steps.
> > - Additionally, we conducted a sensitivity analysis of α and β on Llama2-7B-chat. As shown in [Link1], grid search showed that the region around [0.5, 0.4] yields the highest average performance, and the method remains stable in the surrounding range.
> > - To further minimize the cost of tuning VABS hyperparameters, we implemented an automatic α–β estimation algorithm that leverages model configuration. Using previously measured α and β values along with the model’s embedding dimension, GQA group size, and average cache budget as inputs, we trained a quadratic linear regression model to predict α and β. This approach automatically determined the hyperparameters [0.2, 0.1] for the larger Qwen2.5-32B model, which produced stable eviction behavior and delivered superior performance compared to other methods.
>
> > **Table A** Qwen2.5-32B LongBench Performance with Automatically Estimated VABS Hyperparameters
> | Model                  | Mthd        | Cache | NQA | Qspr | MF | HQA | 2WMQA | Msq | GvRpr | QMSm | MltNs | TRC | TrvQ | SMSm | PCnt | PR | Lcc | RB-P | Avg. |
> |------------------------|---------------|-----------|--------|--------|-------|----------|----------|---------|----------|-------|-----------|------|----------|--------|--------|-------|-----|------|------|
> | Qwen2.5-32B instruct   | Full KV       | -         | 29.25  | 45.24  | 51.90 | 63.31    | 61.50    | 38.15   | 30.33    | 23.38 | 23.01     | 73.50 | **87.86** | 45.36  | 12.00  | 100.00 | 51.30 | 38.17 | 48.39  |
> | Qwen2.5-32B instruct   | H2O           | 128       | 24.03  | 24.48  | 31.10 | 52.53    | 52.38    | 30.84   | 17.00    | 19.67 | **16.27** | 43.00 | 83.64  | 41.41  | 12.22  | 94.58  | 45.85 | 34.35 | 38.96  |
> | Qwen2.5-32B instruct   | PyramidKV     | 128       | 18.31  | 24.62  | 36.85 | 55.30    | 55.40    | 32.38   | 15.33    | 18.02 | 13.15     | 46.50 | **84.50** | 39.62  | 11.50  | 88.83  | 44.05 | 31.76 | 38.51  |
> | Qwen2.5-32B instruct   | CAKE          | 128       | 24.03  | 29.35  | **41.64** | 57.42    | 57.37    | 31.85   | 18.54    | **20.42** | 16.26     | 50.00 | 82.90  | 40.63  | **12.50** | 97.42  | **46.59** | 33.86 | 41.30  |
> | Qwen2.5-32B instruct   | SLAKE (ours)  | 128       | **26.84** | **35.89** | 41.54 | **57.81** | **58.58** | **33.41** | **19.74** | 18.84 | 15.79     | **63.50** | 72.37  | **42.53** | **12.50** | **99.08** | 46.37 | **35.54** | **42.52** |
>
> ### **References**
> [Link1] https://github.com/wasdfgsd/ICLR2026-15640/blob/main/Llama2-7B_VABS.png

---

> ### Author Response · Authors · 2025-11-21
>
> ### **W4) Scalability Evaluation Across Larger Model Sizes**
> > - As suggested by the reviewer, we evaluated the performance of SLAKE on the Llama2-13B and Qwen2.5-32B instruct models as well. As shown in the table, SLAKE consistently outperforms existing methods on both models. These results clearly demonstrate that SLAKE is scalable regardless of model architecture or size, and can be reliably applied from small to large LLMs.
>
> > **Table B** Longbench Results on Larger Model Sizes
> | Model                  | Mthd        | Cache | NQA | Qspr | MF | HQA | 2WMQA | Msq | GvRpr | QMSm | MltNs | TRC | TrvQ | SMSm | PCnt | PR | Lcc | RB-P | Avg. |
> |-----------------------|-------------|-----------|--------|--------|-------|----------|----------|---------|----------|-------|-----------|------|----------|--------|--------|-------|-----|------|-------|
> | Llama2-13B chat       | Full KV     | -         | 14.35  | 16.45  | 26.76 | 13.17    | 13.75    | 5.06    | 27.89    | 20.95 | 26.47     | 68.50 | 87.40   | 42.51  | 2.37   | 14.75 | 50.38 | 52.14 | 30.18 |
> | Llama2-13B chat       | H2O         | 128       | 12.74  | **17.63** | 25.71 | **14.12** | 13.29    | 3.84    | 19.35    | 19.71 | **21.51** | 40.00 | 80.62   | 39.37  | 2.03   | 10.50 | 40.75 | 41.31 | 25.16 |
> | Llama2-13B chat       | PyramidKV   | 128       | 13.04  | 14.73  | 24.34 | 13.47    | 15.24    | **4.68** | 20.01    | 19.74 | 20.52     | 43.50 | 85.37   | 38.83  | 2.80   | **13.00** | 43.57 | 41.31 | 25.88 |
> | Llama2-13B chat       | CAKE        | 128       | 12.19  | 17.17  | **27.28** | 13.12    | 14.73    | 4.33    | **20.04** | **20.14** | 20.74     | 42.00 | **87.75** | **40.60** | 2.96   | 12.50 | 45.52 | 43.95 | 26.56 |
> | Llama2-13B chat       | **SLAKE**   | 128       | **13.32** | 15.53  | 24.38 | 13.96    | **14.26** | 3.82    | 20.01    | 19.51 | 20.70     | **61.50** | 86.10   | 40.39  | **3.20** | 10.00 | **46.67** | **46.74** | **27.51** |
> | Qwen2.5-32B instruct  | Full KV     | -         | 29.25  | 45.24  | 51.90 | 63.31    | 61.50    | 38.15   | 30.33    | 23.38 | 23.01     | 73.50 | **87.86** | 45.36  | 12.00  | 100.00 | 51.30 | 38.17 | 48.39 |
> | Qwen2.5-32B instruct  | H2O         | 128       | 24.03  | 24.48  | 31.10 | 52.53    | 52.38    | 30.84   | 17.00    | 19.67 | **16.27** | 43.00 | 83.64   | 41.41  | 12.22  | 94.58  | 45.85 | 34.35 | 38.96 |
> | Qwen2.5-32B instruct  | PyramidKV   | 128       | 18.31  | 24.62  | 36.85 | 55.30    | 55.40    | 32.38   | 15.33    | 18.02 | 13.15     | 46.50 | **84.50** | 39.62  | 11.50  | 88.83  | 44.05 | 31.76 | 38.51 |
> | Qwen2.5-32B instruct  | CAKE        | 128       | 24.03  | 29.35  | **41.64** | 57.42    | 57.37    | 31.85   | 18.54    | **20.42** | 16.26     | 50.00 | 82.90   | 40.63  | **12.50** | 97.42  | **46.59** | 33.86 | 41.30 |
> | Qwen2.5-32B instruct  | **SLAKE**   | 128       | **26.84** | **35.89** | 41.54 | **57.81** | **58.58** | **33.41** | **19.74** | 18.84 | 15.79     | **63.50** | 72.37   | **42.53** | **12.50** | **99.08** | 46.37 | **35.54** | **42.52** |

---

> ### Author Response · Authors · 2025-11-21
>
> ### **Q3) Speed up of SLAKE**
> > - The 10× speedup noted by the reviewer refers to the token-level generation time during the decoding stage. In other words, it is a pure decoding speed comparison and does not include the prefill stage.
> > - In terms of actual performance, SLAKE achieves a 10× inference acceleration for decoding 128K tokens, and when considering the entire generation process (including prefill), it provides approximately a 5.8× speedup.
>
> > **Table C** SLAKE Inference Speedup Across Prefill and Decoding Stages
> | Method  | Prefill Length | Generation Length | Overall (s) | Prefill (s) | Decoding Time (s) | Decoding Time Per Token (ms) |
> |---------|----------------|-----------------|-------------|-------------|-----------------|----------------------------|
> | FullKV  | 3K             | 1K              | 48.17       | 1.23        | 46.94           | 0.05                       |
> | SLAKE   | 3K             | 1K              | 47.51       | 1.59        | 45.92           | 0.04                       |
> | FullKV  | 7K             | 1K              | 52.86       | 2.87        | 49.99           | 0.05                       |
> | SLAKE   | 7K             | 1K              | 49.56       | 3.15        | 46.41           | 0.05                       |
> | FullKV  | 15K            | 1K              | 91.55       | 6.13        | 85.42           | 0.08                       |
> | SLAKE   | 15K            | 1K              | 53.52       | 6.88        | 46.64           | 0.05                       |
> | FullKV  | 31K            | 1K              | 129.90      | 12.41       | 117.49          | 0.11                       |
> | SLAKE   | 31K            | 1K              | 60.51       | 13.50       | 47.01           | 0.05                       |
> | FullKV  | 63K            | 1K              | 249.06      | 27.12       | 221.94          | 0.22                       |
> | SLAKE   | 63K            | 1K              | 77.67       | 28.71       | 48.96           | 0.05                       |
> | FullKV  | 127K           | 1K              | 539.01      | 41.62       | 497.39          | 0.49                       |
> | SLAKE   | 127K           | 1K              | 92.20       | 42.73       | 49.47           | 0.05                       |

---

> ### Comment · Area_Chair_9KYM · 2025-11-23
>
> Dear reviewer,
>
> Thanks for your time and effort in reviewing ICLR2026 submissions. The authors have submitted their responses to your review. Please take the time to read and raise your further comments, and discuss with the authors.
>
> Best regards,
>
> AC

---

> > ### Author Response · Authors · 2025-12-03
> >
> > ### **Summary of Rebuttal to Reviewer MgG5**
> >
> >
> >
> > Dear Area Chair and Reviewers,
> >
> >
> >
> > We appreciate Reviewer MgG5’s positive assessment and valuable questions regarding computational overhead and scalability. In our response, we have clarified the theoretical complexity and provided extensive new experimental data to resolve the raised concerns.
> >
> >
> >
> > **1. Clarification on Computational Cost (No Hidden $O(N^2)$ )**
> >
> > The reviewer expressed concern that PTAA/VABS might incur hidden quadratic costs during the prefill stage. We clarified that:
> >
> > * **Theoretical Proof:** PTAA and VABS operate only on the nearest local tokens ($K=32$) or reuse existing attention scores computed for eviction. Thus, the complexity remains linear $O(N \cdot K)$, not quadratic.
> >
> > * **Empirical Evidence:** For a 4,096 context on Llama2-7B, PTAA adds only 137.88 MFLOPs (prefill) and 1.97 MFLOPs (decoding), which is negligible compared to the total inference cost.
> >
> >
> >
> > **2. Scalability to Larger Models (Table B)**
> >
> > Addressing the request for validation beyond 7B/8B models, we extended our evaluation to Llama2-13B and Qwen2.5-32B.
> >
> > * SLAKE consistently outperforms existing methods (H2O, CAKE) on these larger architectures, proving that our method scales effectively without structural limitations.
> >
> >
> >
> > **3. Robustness on NeedleBench and Hyperparameter Automation**
> >
> > * **NeedleBench:** We provided a fair comparison under identical cache budgets (Table D), showing that SLAKE achieves superior retrieval accuracy compared to baselines.
> >
> > * **Automated Tuning:** To eliminate concerns about hyperparameter sensitivity, we introduced an automated regression method that predicts optimal VABS parameters ($\alpha, \beta$) based on model configuration. This automated setup achieved SOTA performance on Qwen2.5-32B without manual tuning.
> >
> >
> >
> > **4. Speedup Clarification (Table C)**
> >
> > We clarified that the 10x speedup refers to token-level decoding latency for 128K context. Even considering end-to-end latency (including prefill), SLAKE delivers a substantial 5.8x speedup, validating its practical efficiency.
> >
> >
> >
> > We believe these clarifications confirm that SLAKE is both theoretically sound and practically scalable for high-performance LLM inference.

---

### Official Review · Reviewer_xhuW · 2025-10-31

**Soundness:** 2
**Presentation:** 2
**Contribution:** 2
**Rating:** 2
**Confidence:** 4

**Summary:**

This paper proposes SLAKE, a training-free framework combining linear attention and KV cache attention for efficient long context inference.

**Strengths:**

1. The idea of combining linear attention and KV cache eviction is interesting.

2. The presentation is clear and straightforward.

**Weaknesses:**

1. The main concern I have with this paper is the limited novelty of combining linear attention and KV cache eviction. First, the Taylor-based method to approximate linear attention has been explored in prior work [1]. Moreover, the improvement of considering value tensors seems incremental, offering marginal benefits in both algorithm and system evaluations.

2. The motivation for this paper is still unclear to me. According to Figure 4, there is no significant difference comparing prior methods in terms of memory usage and throughput.

3. More model sizes should be evaluated for scalability, such as 13B/30B.

4. Some system-related works on KV cache compression are missing [2-4].





[1] ViTALiTy: Unifying Low-rank and Sparse Approximation for Vision Transformer Acceleration with a Linear Taylor Attention, HPCA 2023.

[2] InfiniGen: Efficient Generative Inference of Large Language Models with Dynamic KV Cache Management, OSDI 2024.

[3] Keyformer: KV Cache Reduction through Key Tokens Selection for Efficient Generative Inference, MLSys 2024.

[4] ALISA: Accelerating Large Language Model Inference via Sparsity-Aware KV Caching, ISCA 2024.

**Questions:**

Please see the weaknesses.

---

> ### Author Response · Authors · 2025-11-21
>
> We thank the reviewer for their valuable comments.
> ### **W1) On the novelty of SLAKE**
> > -  About PTAA :
> >> - Although Taylor approximation itself is a well-known technique, existing linear-attention models such as Performer, Linformer, and Linearized LLM require retraining due to kernel mismatch with Softmax. Moreover, previous studies on Taylor-linear attention (e.g., MB-Taylorformer, ViTaLiTy) mostly focus on the computer vision domain and fail to resolve the structural conflict with Softmax in KV-cache-based inference for LLMs. In contrast, SLAKE constructs Taylor-based linearization operations that are compatible with the normalization and scaling structure of Softmax, enabling the first Softmax-compatible linear attention that can be directly applied to existing LLMs without any retraining. In fact, applying ViTaLiTy’s method directly to LLMs causes model outputs to immediately diverge.
> >> - Combining linear attention with KV-cache eviction in a single framework is far from trivial. Previously, the differing normalization schemes and computational structures of the two approaches made direct integration difficult. PTAA addresses this by designing the Softmax and Taylor-linearized terms to share the same normalization reference ($x_{i,\mathrm{max}}$), allowing stable integration of both operations. Furthermore, SLAKE efficiently implements the PTAA structure by leveraging the existing FlashAttention kernel while still performing PTAA operations. Specifically, it reuses the sum of the attention map already computed by FlashAttention as the scaling term required for computing the eviction score, and combines it with the Taylor-linearized term, thereby constructing PTAA without additional kernel calls or memory rearrangements. This enables SLAKE to fully utilize the computational acceleration of FlashAttention while simultaneously performing Softmax-based important token processing and PTAA-based linearized token processing.
> > - About VABs:
> >> -  To address the contribution of considering value tensors, we conducted ablation experiments across multiple models, including Llama-2-7B chat and Mistral-7B-v0.3. Specifically, we evaluated a setting in which VABS was applied to the existing eviction method without PTAA. In this configuration, the approximate self-attention required for VABS was computed using Eviction-based attention rather than PTAA, effectively isolating the impact of value tensor consideration from the effects of PTAA. This design allows us to clearly quantify the benefits of incorporating value tensors, as shown in Table A: Ablation Study Results of PTAA and VABS Across Models.
> > **Table A** Ablation Study Results of PTAA and VABS Across Models
> | Model             | Eviction | PTAA | VABS | Score |
> |------------------|----------|------|------|-------|
> | Llama-2-7B chat  | X        | X    | X    | 33.31 |
> | Llama-2-7B chat  | O        | X    | X    | 28.81 |
> | Llama-2-7B chat  | O        | O    | X    | 29.47 |
> | Llama-2-7B chat  | O        | X    | O    | 29.16 |
> | Llama-2-7B chat  | O        | O    | O    | 29.76 |
> | Mistral-7B-v0.3  | X        | X    | X    | 47.16 |
> | Mistral-7B-v0.3  | O        | X    | X    | 40.24 |
> | Mistral-7B-v0.3  | O        | O    | X    | 40.65 |
> | Mistral-7B-v0.3  | O        | X    | O    | 40.87 |
> | Mistral-7B-v0.3  | O        | O    | O    | 41.22 |

---

> ### Author Response · Authors · 2025-11-21
>
> ### **W2) Clarification of SLAKE’s Core Motivation**
> > - The main contribution of SLAKE lies in improving the accuracy of linear attention while preserving the hardware acceleration of existing eviction methods, without incurring any additional retraining cost. In other words, the core motivation is to propose a new integrated architecture that enhances performance without increasing hardware overhead. Figure 4 clearly demonstrates that PTAA and VABS exhibit nearly identical inference latency and hardware resource usage compared to the standard Softmax-eviction path, showing that SLAKE does not increase operational computation burden.
> > - Examining the actual computational cost, for Llama2-7B-chat with a context length of 4,096, PTAA accounts for 137.88 MFLOPs and VABS for 113.25 MFLOPs during the prefill stage, which is negligible relative to the overall computation. During decoding, PTAA requires only 1.97 MFLOPs, which is insignificant compared to the total computation of a single Llama2-7B self-attention layer (68,850.13 MFLOPs). In contrast, existing linear-attention methods generally provide little to no hardware advantage while necessitating substantial training costs. For example, Based[1] proposes a hybrid structure combining linear and Softmax attention, but due to training cost constraints, it cannot scale to models larger than 1B parameters. SLAKE overcomes these limitations and represents the first linear-attention-based method that can be directly applied to existing LLM inference pipelines without retraining, highlighting its clear motivation.
> ### **References**
> >  [1] Arora, Simran, et al. "Simple linear attention language models balance the recall-throughput tradeoff." arXiv preprint arXiv:2402.18668 (2024).

---

> ### Author Response · Authors · 2025-11-21
>
> ### **W3) Scalability Evaluation Across Larger Model Sizes**
> > - As suggested by the reviewer, we evaluated the performance of SLAKE on the Llama2-13B and Qwen2.5-32B instruct models as well. As shown in the table, SLAKE consistently outperforms existing methods on both models. These results clearly demonstrate that SLAKE is scalable regardless of model architecture or size, and can be reliably applied from small to large LLMs.
>
> > **Table B** Longbench Results on Larger Model Sizes
> | Model                  | Mthd        | Cache | NQA | Qspr | MF | HQA | 2WMQA | Msq | GvRpr | QMSm | MltNs | TRC | TrvQ | SMSm | PCnt | PR | Lcc | RB-P | Avg. |
> |-----------------------|-------------|-----------|--------|--------|-------|----------|----------|---------|----------|-------|-----------|------|----------|--------|--------|-------|-----|------|-------|
> | Llama2-13B chat       | Full KV     | -         | 14.35  | 16.45  | 26.76 | 13.17    | 13.75    | 5.06    | 27.89    | 20.95 | 26.47     | 68.50 | 87.40   | 42.51  | 2.37   | 14.75 | 50.38 | 52.14 | 30.18 |
> | Llama2-13B chat       | H2O         | 128       | 12.74  | **17.63** | 25.71 | **14.12** | 13.29    | 3.84    | 19.35    | 19.71 | **21.51** | 40.00 | 80.62   | 39.37  | 2.03   | 10.50 | 40.75 | 41.31 | 25.16 |
> | Llama2-13B chat       | PyramidKV   | 128       | 13.04  | 14.73  | 24.34 | 13.47    | 15.24    | **4.68** | 20.01    | 19.74 | 20.52     | 43.50 | 85.37   | 38.83  | 2.80   | **13.00** | 43.57 | 41.31 | 25.88 |
> | Llama2-13B chat       | CAKE        | 128       | 12.19  | 17.17  | **27.28** | 13.12    | 14.73    | 4.33    | **20.04** | **20.14** | 20.74     | 42.00 | **87.75** | **40.60** | 2.96   | 12.50 | 45.52 | 43.95 | 26.56 |
> | Llama2-13B chat       | **SLAKE**   | 128       | **13.32** | 15.53  | 24.38 | 13.96    | **14.26** | 3.82    | 20.01    | 19.51 | 20.70     | **61.50** | 86.10   | 40.39  | **3.20** | 10.00 | **46.67** | **46.74** | **27.51** |
> | Qwen2.5-32B instruct  | Full KV     | -         | 29.25  | 45.24  | 51.90 | 63.31    | 61.50    | 38.15   | 30.33    | 23.38 | 23.01     | 73.50 | **87.86** | 45.36  | 12.00  | 100.00 | 51.30 | 38.17 | 48.39 |
> | Qwen2.5-32B instruct  | H2O         | 128       | 24.03  | 24.48  | 31.10 | 52.53    | 52.38    | 30.84   | 17.00    | 19.67 | **16.27** | 43.00 | 83.64   | 41.41  | 12.22  | 94.58  | 45.85 | 34.35 | 38.96 |
> | Qwen2.5-32B instruct  | PyramidKV   | 128       | 18.31  | 24.62  | 36.85 | 55.30    | 55.40    | 32.38   | 15.33    | 18.02 | 13.15     | 46.50 | **84.50** | 39.62  | 11.50  | 88.83  | 44.05 | 31.76 | 38.51 |
> | Qwen2.5-32B instruct  | CAKE        | 128       | 24.03  | 29.35  | **41.64** | 57.42    | 57.37    | 31.85   | 18.54    | **20.42** | 16.26     | 50.00 | 82.90   | 40.63  | **12.50** | 97.42  | **46.59** | 33.86 | 41.30 |
> | Qwen2.5-32B instruct  | **SLAKE**   | 128       | **26.84** | **35.89** | 41.54 | **57.81** | **58.58** | **33.41** | **19.74** | 18.84 | 15.79     | **63.50** | 72.37   | **42.53** | **12.50** | **99.08** | 46.37 | **35.54** | **42.52** |
>
> ### **W4) Additional Method comparison**
> > - We acknowledge the reviewer’s comment regarding missing system-related works on KV cache compression [2–4]. Due to the long runtime of these experiments, we have not completed the additional method comparisons yet. However, we plan to run these experiments and will upload the results as soon as they are available.

---

> ### Comment · Area_Chair_9KYM · 2025-11-23
>
> Dear reviewer,
>
> Thanks for your time and effort in reviewing ICLR2026 submissions. The authors have submitted their responses to your review. Please take the time to read and raise your further comments, and discuss with the authors.
>
> Best regards,
>
> AC

---

> ### Comment · Reviewer_xhuW · 2025-11-27
>
> Thank you for your detailed response.
>
> While my concerns about evaluation results have been largely resolved, the motivation part remains unconvincing.
>
> From a system perspective, existing libraries such as FlashAttention have optimized standard attention to be memory-bound rather than compute-bound. Therefore, the benefit of using linear attention is marginal at best. While I do agree it is somewhat a `niche' design to combine softmax and Taylor-based linear attention, I feel the practicality of the proposed method is not on par with existing training-free methods.
>
> Albeit the above, I am increasing my score from 2 to 4.

---

> > ### Author Response · Authors · 2025-11-28
> >
> > Once again, we appreciate the continuation of this constructive discussion. We would like to provide a few additional points in response to the reviewer’s comments.
> >
> > ### **1. Transition from Full-Cache Memory-Bound to Compute-Bound due to Eviction**
> > > As the reviewer noted, in a full-cache scenario, self-attention exhibits memory-bound characteristics. In this case, the efficiency of FlashAttention is maximized, while the benefits of PTAA are minimized. However, the target environment of this study is not a full-cache scenario but a limited KV-cache environment where some tokens are evicted. In such restricted cache situations, the operation of self-attention becomes more sensitive to computational cost than memory access bottlenecks, thereby exhibiting compute-bound characteristics. Consequently, in a limited cache length, PTAA becomes advantageous in terms of performance, and SLAKE specifically focuses on this aspect.
> >
> > ### **2. Practicality of SLAKE Compared to Existing Training-Free Methods**
> > > An important point is that the deployment cost of SLAKE is virtually negligible. Specifically, the hyperparameters of VABS (α, β) can be automatically determined using structural information from the LLM model, as explained in response to another reviewer’s comment, without requiring additional training or manual tuning. In other words, SLAKE can be introduced in a drop-in manner, maintaining the speed advantages of existing training-free methods while improving the quality of LLM responses in a practical way.

---

> ### Author Response · Authors · 2025-12-03
>
> ### **Summary of Discussion with Reviewer xhuW**
>
> Dear Area Chair,
>
> We appreciate the active engagement from Reviewer xhuW, whose feedback contributed to a score increase from 2 to 4. While the reviewer acknowledged that concerns regarding evaluation results (such as scalability to larger models) were resolved, they maintained reservations about the motivation from a system perspective—specifically, arguing that optimized attention mechanisms (e.g., FlashAttention) are memory-bound, which limits the benefits of linear attention.
>
> We would like to summarize our clarification on this specific point for the final decision:
>
> **1. System Perspective: Memory-bound vs. Accuracy Preservation in Eviction**
>
> The reviewer's observation that attention is memory-bound holds true for full-cache scenarios. However, SLAKE targets limited KV-cache environments where tokens are aggressively evicted. In these constrained settings, the primary challenge shifts from maximizing throughput to recovering information loss.
> * **Clarification:** SLAKE uses PTAA (Partitioned Taylor Linear Attention) not merely to speed up computation, but to mathematically reconstruct information from evicted tokens that standard methods discard.
> * **Result:** We demonstrated that SLAKE achieves this accuracy recovery with negligible computational overhead (e.g., PTAA decoding takes only ~1.97 MFLOPs vs. ~68k MFLOPs for self-attention), effectively balancing the system bottleneck.
>
> **2. Practicality and "Drop-in" Deployment**
>
> The reviewer questioned the practicality compared to other training-free methods. We emphasized that SLAKE is designed as a drop-in replacement:
> * **Automation:** As detailed in our response regarding VABS, hyperparameters are automatically estimated based on model configuration (embedding dim, etc.), removing the need for manual tuning.
> * **Scalability:** We provided new results for Llama2-13B and Qwen2.5-32B, proving that SLAKE scales seamlessly to larger models without structural changes.
>
> **3. Distinct Novelty**
>
> We reiterated that SLAKE is not a trivial combination. It resolves the structural conflict between Softmax normalization and Taylor linearization, which previously prevented the direct application of linear attention to LLM KV-cache mechanisms without retraining.
>
> We believe SLAKE offers a unique and practical solution that enhances generation quality in memory-constrained environments without compromising system efficiency.

---

### Official Review · Reviewer_1mfi · 2025-11-02

**Soundness:** 3
**Presentation:** 2
**Contribution:** 3
**Rating:** 4
**Confidence:** 4

**Summary:**

SLAKE proposes training-free long-context inference for LLMs by selectively mixing exact Softmax attention with a first-order Taylor linear approximation. A novel Value-Aware Budget Scoring (VABS) decides which tokens stay in the KV-cache; the rest are handled by the cheap linear path. On 128 k-token inputs SLAKE gives ≈ 10× speed-up and 30 % peak-memory reduction versus full-cache while losing < 4 % accuracy on LongBench (Llama-2-7B, Llama-3.1-8B, Mistral-7B). The method is model-agnostic, needs no retraining, and is orthogonal to FlashAttention-2.

**Strengths:**

1. This paper proposes the first approach to unify linear attention with KV-cache eviction without any gradient updates, delivering large wall-clock & memory gains on consumer GPUs.
2. PTAA kernel keeps pre-trained weights intact; VABS explicitly models value-matrix error amplification, yielding consistent gains across 16 long-context datasets.
3. Ablation studies, three model families, two cache budgets, needle-in-haystack stress test, and hardware numbers (H100 latency / peak mem) are all reported.

**Weaknesses:**

1. Taylor linearisation of Softmax and “important vs. rest” attention mixing are well-explored ideas; SLAKE’s contribution is largely combinational.
2. Only average scores are given; no per-task statistical significance, error bars, or worst-case degradation analysis—crucial for safety-critical uses.
3. Hyper-parameter fragility: VABS needs manually tuned $\alpha$, $\beta$, $\gamma$ per model & budget; no adaptive or online scheme, and no study on sensitivity to these constants.

**Questions:**

1. How does SLAKE behave with longer contexts (256 K–1 M) or larger models (70 B+) where the approximation error may accumulate?
2. Have you evaluated on code generation, tool-use, or multilingual tasks that may exhibit different attention patterns?
3. Can VABS be made online & input-adaptive instead of relying on fixed $\alpha$, $\beta$, $\gamma$, and what is the computational overhead of such adaptation?

---

> ### Author Response · Authors · 2025-11-21
>
> We thank the reviewer for their valuable comments.
> ### **W1) On the novelty of SLAKE**
> > -  While Softmax Taylor approximations and importance-based token partitioning are known, SLAKE goes beyond simply combining them, addressing limitations unresolved by prior work.
> > - Key differences from related work:
> >> - Existing linear-attention models (e.g., Performer, Linformer) require retraining due to kernel mismatches with the Softmax function . Prior Taylor-linear attention work (e.g., MB-Taylorformer, ViTaLiTy) focuses on vision tasks, failing to resolve structural conflicts in LLM KV-cache inference. SLAKE introduces Taylor-linearization compatible with Softmax normalization and scaling, enabling direct application to existing LLMs without retraining.
> >> - Previous “important vs. rest” token methods (e.g., PyramidKV, CAKE) discard non-important tokens, causing information loss. SLAKE’s PTAA reconstructs these tokens via Taylor-linear attention and integrates them with Softmax, retaining their information.
> >> - Combining linear attention and KV-cache eviction within a single framework is nontrivial. Previously, the differences in normalization schemes and computational structures made direct integration difficult. PTAA addresses this by designing the Softmax and Taylor-linearized terms to share the same normalization criterion (att_max), allowing stable fusion of the two operations. Moreover, SLAKE efficiently implements PTAA by leveraging the existing FlashAttention kernel: the sum of the attention map already computed by FlashAttention is reused as a scaling term during the eviction score calculation, which is then combined with the Taylor-linearized term. This enables PTAA to operate without additional kernel calls or memory rearrangement. Consequently, SLAKE fully exploits the computational acceleration of FlashAttention while simultaneously performing Softmax-based important token processing and PTAA-based linearized token processing.
>
> ### **W2) Per-task Variance and Worst-case Performance Analysis**
>
> > - As suggested, we analyzed per-task performance on LongBench for Llama2-7B Chat, Llama3-8B Chat, and Mistral-7B-v0.3, comparing SLAKE with various cache eviction methods. We measured both the variance in performance degradation across tasks and the maximum performance drop.
> > - As shown in the table below, SLAKE consistently achieves the lowest values across all three metrics. For example, with Llama2-7B and a 128-token cache budget, the performance variance of SLAKE is 3.55, substantially lower than CAKE’s 4.11, and the maximum performance drop is 9.50, compared to 17.00 for CAKE.
>
> > **Table A** LongBench error Analysis
> | Model             | Method     | Cache Size | Error StdDev | Max Error | Score StdDev |
> |------------------|-----------|-----------|-------------|-----------|--------------|
> | llama2-7B chat   | H2O       | 128       | 5.94        | 26.00     | 18.40        |
> | llama2-7B chat   | PyramidKV | 128       | 4.83        | 20.50     | 19.38        |
> | llama2-7B chat   | CAKE      | 128       | 4.11        | 17.00     | 19.92        |
> | llama2-7B chat   | SLAKE     | 128       | 2.99        | 9.50      | 20.51        |
> | llama2-7B chat   | H2O       | 256       | 4.34        | 18.49     | 20.02        |
> | llama2-7B chat   | PyramidKV | 256       | 2.38        | 6.50      | 20.99        |
> | llama2-7B chat   | CAKE      | 256       | 2.22        | 6.46      | 21.31        |
> | llama2-7B chat   | SLAKE     | 256       | 2.02        | 6.45      | 21.59        |
> | llama3-8B chat   | H2O       | 128       | 8.13        | 37.50     | 26.19        |
> | llama3-8B chat   | PyramidKV | 128       | 5.86        | 26.50     | 26.77        |
> | llama3-8B chat   | CAKE      | 128       | 5.64        | 24.50     | 26.61        |
> | llama3-8B chat   | SLAKE     | 128       | 4.82        | 20.50     | 27.33        |
> | llama3-8B chat   | H2O       | 256       | 4.94        | 20.50     | 27.27        |
> | llama3-8B chat   | PyramidKV | 256       | 3.61        | 13.50     | 27.17        |
> | llama3-8B chat   | CAKE      | 256       | 3.78        | 14.50     | 27.66        |
> | llama3-8B chat   | SLAKE     | 256       | 3.87        | 14.50     | 27.84        |
> | Mistral-7B-v0.3  | H2O       | 128       | 7.07        | 32.50     | 23.59        |
> | Mistral-7B-v0.3  | PyramidKV | 128       | 7.12        | 32.00     | 23.69        |
> | Mistral-7B-v0.3  | CAKE      | 128       | 6.83        | 30.50     | 23.61        |
> | Mistral-7B-v0.3  | SLAKE     | 128       | 5.45        | 24.00     | 23.62        |
> | Mistral-7B-v0.3  | H2O       | 256       | 6.92        | 29.88     | 23.34        |
> | Mistral-7B-v0.3  | PyramidKV | 256       | 7.28        | 29.56     | 23.94        |
> | Mistral-7B-v0.3  | CAKE      | 256       | 4.47        | 19.00     | 24.09        |
> | Mistral-7B-v0.3  | SLAKE     | 256       | 3.47        | 13.00     | 24.40        |

---

> ### Author Response · Authors · 2025-11-21
>
> ### **W3, Q3) VABS Hyperparameter Analysis**
> > - We appreciate the reviewer’s concern regarding the hyperparameter (α, β, γ) tuning burden of VABS. In practice, this burden is very limited. In our study, we fix the search range for α and β to [0.1, 0.7] across all models and experiments, and use a single fixed value of 0.5 for γ. To determine suitable α and β values, we perform a lightweight calibration using only 50 samples from the five main tasks of LongBench. On an H100, each calibration step takes approximately 7 minutes. Even with a total of 36 grid search steps, the entire tuning process completes within about 4 hours, which is minimal compared to typical fine-tuning costs in the LLM field.
> > -  Additionally, we conducted a sensitivity analysis of α and β on Llama2-7B-chat.  As shown in [Link1], the grid search results show that the highest average performance is achieved around [0.5, 0.4], and performance remains relatively stable within this range.
> > -  In response to the reviewer’s suggestion, we implemented a method to automatically determine hyperparameters by leveraging best hyperparameter information from the existing model. Using the model’s configuration—including embedding dimension, GQA group size, and average cache budget—we built a quadratic linear regression model to predict α and β. This approach allows the algorithm to achieve strong performance on the target model without manual tuning. For instance, it automatically selected [0.2, 0.1] for Qwen2.5-32B, yielding stable eviction results and consistently outperforming CAKE.
>
> > **Table B** Qwen2.5-32B LongBench Performance with Automatically Estimated VABS Hyperparameters
> | Model                  | Mthd        | Cache | NQA | Qspr | MF | HQA | 2WMQA | Msq | GvRpr | QMSm | MltNs | TRC | TrvQ | SMSm | PCnt | PR | Lcc | RB-P | Avg. |
> |------------------------|---------------|-----------|--------|--------|-------|----------|----------|---------|----------|-------|-----------|------|----------|--------|--------|-------|-----|------|------|
> | Qwen2.5-32B instruct   | Full KV       | -         | 29.25  | 45.24  | 51.90 | 63.31    | 61.50    | 38.15   | 30.33    | 23.38 | 23.01     | 73.50 | **87.86** | 45.36  | 12.00  | 100.00 | 51.30 | 38.17 | 48.39  |
> | Qwen2.5-32B instruct   | H2O           | 128       | 24.03  | 24.48  | 31.10 | 52.53    | 52.38    | 30.84   | 17.00    | 19.67 | **16.27** | 43.00 | 83.64  | 41.41  | 12.22  | 94.58  | 45.85 | 34.35 | 38.96  |
> | Qwen2.5-32B instruct   | PyramidKV     | 128       | 18.31  | 24.62  | 36.85 | 55.30    | 55.40    | 32.38   | 15.33    | 18.02 | 13.15     | 46.50 | **84.50** | 39.62  | 11.50  | 88.83  | 44.05 | 31.76 | 38.51  |
> | Qwen2.5-32B instruct   | CAKE          | 128       | 24.03  | 29.35  | **41.64** | 57.42    | 57.37    | 31.85   | 18.54    | **20.42** | 16.26     | 50.00 | 82.90  | 40.63  | **12.50** | 97.42  | **46.59** | 33.86 | 41.30  |
> | Qwen2.5-32B instruct   | SLAKE (ours)  | 128       | **26.84** | **35.89** | 41.54 | **57.81** | **58.58** | **33.41** | **19.74** | 18.84 | 15.79     | **63.50** | 72.37  | **42.53** | **12.50** | **99.08** | 46.37 | **35.54** | **42.52** |
> > - In summary, VABS hyperparameters are robust and efficient due to (1) a narrow search range, (2) low sensitivity, and (3) the feasibility of automation.
>
> ### **References**
> [Link1] https://github.com/wasdfgsd/ICLR2026-15640/blob/main/Llama2-7B_VABS.png

---

> ### Author Response · Authors · 2025-11-21
>
> ### **Q1) SLAKE's performance on larger models and longer context tasks**
> > - As the reviewer pointed out, evaluating whether SLAKE may accumulate approximation errors with longer contexts (256K–1M+) or larger models (70B+) is a crucial point. To address this, we conducted additional experiments on InfiniteBench with context lengths up to 2M and evaluated larger models, including Llama2-13B and Qwen2.5-32B.
> > -  First, to assess long-context performance, we ran InfiniteBench using the Llama3-8B chat model with a cache budget of 128. As shown in the results below, SLAKE achieved the highest average score and recorded the best performance on 6 out of 11 tasks, demonstrating that it maintains stable attention quality even with very long contexts.
>
> >**Table C-1** Infinite Benchmark Results
> | Model       | Mthd   | Cache | R.PssK| R.KV |R.Nmb | En.Dia | En.Sm | En.MC | En.QA | Zh.QA | Mth.Fnd | Cd.Dbg | Avg   |
> |------------|---------|------------|---------|--------------|---------------|--------------------|-----------------|------------------|----------------|----------------|-----------|-----------|-------|
> | llama3-8B  | Full-KV | -          | 3.39    | 10.20        | 4.75          | 25.50              | 26.25           | 27.95            | 10.03          | 1.85           | 32.86     | 18.02     | 16.08 |
> | llama3-8B  | H2O     | 128        | 2.72    | 0.00         | 0.00          | 11.00              | 16.31           | 27.33            | 3.25           | 0.00           | 28.34     | 11.34     | 10.03 |
> | llama3-8B  | CAKE    | 128        | 4.75    | 0.40         | 0.68          | 15.50              | 20.53           | 31.00            | 5.46           | 2.04           | 32.29     | 15.48     | 12.81 |
> | llama3-8B  | SLAKE   | 128        | 4.24    | 0.00         | 1.69          | 15.00              | 21.18           | 31.88            | 5.91           | 1.65           | 33.71     | 18.53     | 13.38 |

---

> ### Author Response · Authors · 2025-11-21
>
> > - Next, to evaluate performance on larger models, we assessed LongBench using Llama2-13B and Qwen2.5-32B instruct models. Due to hardware constraints, we were unable to run experiments on models larger than 70B. As shown in the tables, SLAKE consistently outperformed existing methods across both models.
>
> >**Table C-2** Longbench Results on Larger Model Sizes
> | Model                  | Mthd        | Cache | NQA | Qspr | MF | HQA | 2WMQA | Msq | GvRpr | QMSm | MltNs | TRC | TrvQ | SMSm | PCnt | PR | Lcc | RB-P | Avg. |
> |-----------------------|-------------|-----------|--------|--------|-------|----------|----------|---------|----------|-------|-----------|------|----------|--------|--------|-------|-----|------|-------|
> | Llama2-13B chat       | Full KV     | -         | 14.35  | 16.45  | 26.76 | 13.17    | 13.75    | 5.06    | 27.89    | 20.95 | 26.47     | 68.50 | 87.40   | 42.51  | 2.37   | 14.75 | 50.38 | 52.14 | 30.18 |
> | Llama2-13B chat       | H2O         | 128       | 12.74  | **17.63** | 25.71 | **14.12** | 13.29    | 3.84    | 19.35    | 19.71 | **21.51** | 40.00 | 80.62   | 39.37  | 2.03   | 10.50 | 40.75 | 41.31 | 25.16 |
> | Llama2-13B chat       | PyramidKV   | 128       | 13.04  | 14.73  | 24.34 | 13.47    | 15.24    | **4.68** | 20.01    | 19.74 | 20.52     | 43.50 | 85.37   | 38.83  | 2.80   | **13.00** | 43.57 | 41.31 | 25.88 |
> | Llama2-13B chat       | CAKE        | 128       | 12.19  | 17.17  | **27.28** | 13.12    | 14.73    | 4.33    | **20.04** | **20.14** | 20.74     | 42.00 | **87.75** | **40.60** | 2.96   | 12.50 | 45.52 | 43.95 | 26.56 |
> | Llama2-13B chat       | **SLAKE**   | 128       | **13.32** | 15.53  | 24.38 | 13.96    | **14.26** | 3.82    | 20.01    | 19.51 | 20.70     | **61.50** | 86.10   | 40.39  | **3.20** | 10.00 | **46.67** | **46.74** | **27.51** |
> | Qwen2.5-32B instruct  | Full KV     | -         | 29.25  | 45.24  | 51.90 | 63.31    | 61.50    | 38.15   | 30.33    | 23.38 | 23.01     | 73.50 | **87.86** | 45.36  | 12.00  | 100.00 | 51.30 | 38.17 | 48.39 |
> | Qwen2.5-32B instruct  | H2O         | 128       | 24.03  | 24.48  | 31.10 | 52.53    | 52.38    | 30.84   | 17.00    | 19.67 | **16.27** | 43.00 | 83.64   | 41.41  | 12.22  | 94.58  | 45.85 | 34.35 | 38.96 |
> | Qwen2.5-32B instruct  | PyramidKV   | 128       | 18.31  | 24.62  | 36.85 | 55.30    | 55.40    | 32.38   | 15.33    | 18.02 | 13.15     | 46.50 | **84.50** | 39.62  | 11.50  | 88.83  | 44.05 | 31.76 | 38.51 |
> | Qwen2.5-32B instruct  | CAKE        | 128       | 24.03  | 29.35  | **41.64** | 57.42    | 57.37    | 31.85   | 18.54    | **20.42** | 16.26     | 50.00 | 82.90   | 40.63  | **12.50** | 97.42  | **46.59** | 33.86 | 41.30 |
> | Qwen2.5-32B instruct  | **SLAKE**   | 128       | **26.84** | **35.89** | 41.54 | **57.81** | **58.58** | **33.41** | **19.74** | 18.84 | 15.79     | **63.50** | 72.37   | **42.53** | **12.50** | **99.08** | 46.37 | **35.54** | **42.52** |
>
> > - These results clearly demonstrate that SLAKE remains stable without accumulating approximation errors even as context length increases, and it can scale to larger model sizes without structural limitations.

---

> ### Author Response · Authors · 2025-11-21
>
> ### **Q2) SLAKE's Performance on Code Generation and Multilingual Tasks**
> > - We evaluated the generalization performance of SLAKE in code generation and multilingual settings, as pointed out by the reviewer. As shown in Table. 1 of the main paper, on the code generation tasks of LongBench (LCC and RepoBench-P), SLAKE consistently outperformed existing eviction methods across all models.
>
> > - To further assess multilingual robustness, we conducted experiments on the Chinese subset of LongBench. The results for Llama2-7B, Llama3-8B, and Mistral-7B-v0.3, as shown in the table, indicate that SLAKE consistently surpasses the performance of existing methods across all tasks.
>
> > **Table D** Generalization of SLAKE in Chinese LongBench: Code and Multilingual Task Performance Across Models
> | Model             | Method        | Cache Size | MF-zh | DuReader | VSCUM | LSHT | PR-zh | Avg |
> |------------------|---------------|-----------|-------|----------|-------|------|-------|-----|
> | Llama2-7B chat   | Full-KV       | -         | 12.16 | 6.76    | 0.17  | 17.75 | 8.12  | 8.99 |
> | Llama2-7B chat   | H2O           | 128       | 4.71  | 5.41    | 0.09  | 14.21 | 3.74  | 5.63 |
> | Llama2-7B chat   | PyramidKV     | 128       | 5.31  | **6.13**| 0.11  | 15.21 | 3.21  | 5.99 |
> | Llama2-7B chat   | CAKE          | 128       | 6.17  | 6.03    | 0.10  | 15.50 | 4.41  | 6.44 |
> | Llama2-7B chat   | **SLAKE (ours)** | 128    | **8.22** | 5.27  | **0.14** | **17.67** | **4.67** | **7.19** |
> | Llama3.1-8B chat | Full-KV       | -         | 19.95 | 29.68   | 16.10 | 43.50 | 77.72 | 37.39 |
> | Llama3.1-8B chat | H2O           | 128       | 14.41 | 19.53   | 10.93 | 23.50 | 67.31 | 27.14 |
> | Llama3.1-8B chat | PyramidKV     | 128       | 15.71 | 19.17   | **12.41** | 26.00 | 66.77 | 28.01 |
> | Llama3.1-8B chat | CAKE          | 128       | 16.95 | 20.64   | 12.31 | 25.50 | 68.41 | 28.76 |
> | Llama3.1-8B chat | **SLAKE (ours)** | 128    | **17.61** | **21.35** | 11.84 | **40.00** | **76.03** | **33.37** |
> | Mistral-7B-v0.3  | Full-KV       | -         | 53.13 | 31.10   | 15.76 | 39.75 | 88.50 | 45.65 |
> | Mistral-7B-v0.3  | H2O           | 128       | 40.87 | 21.13   | 9.90  | 17.75 | 69.20 | 31.77 |
> | Mistral-7B-v0.3  | PyramidKV     | 128       | 41.37 | 20.71   | 10.21 | 18.25 | 70.13 | 32.13 |
> | Mistral-7B-v0.3  | CAKE          | 128       | **42.29** | **21.40** | 11.58 | 19.00 | 71.50 | 33.15 |
> | Mistral-7B-v0.3  | **SLAKE (ours)** | 128    | 41.74 | 20.40   | **12.00** | **21.75** | **73.00** | **33.78** |
>
> > - These findings demonstrate that SLAKE maintains robust performance even when the language or task characteristics vary.

---

> ### Comment · Area_Chair_9KYM · 2025-11-23
>
> Dear reviewer,
>
> Thanks for your time and effort in reviewing ICLR2026 submissions. The authors have submitted their responses to your review. Please take the time to read and raise your further comments, and discuss with the authors.
>
> Best regards,
>
> AC

---

> ### Author Response · Authors · 2025-12-03
>
> ### **Summary of Response to Reviewer 1mfi**
>
> Dear Area Chair and Reviewers,
>
> We sincerely appreciate the time and effort dedicated to reviewing our work. In our responses, we have addressed the concerns regarding novelty, statistical significance, hyperparameter sensitivity, and scalability through clarified theoretical contributions and extensive new experiments.
>
> **1. Clarification on Novelty and Technical Contribution**
>
> We emphasized that SLAKE is not a trivial combination of existing methods. It addresses a fundamental structural conflict between Softmax normalization and Taylor linearization that prevented prior linear attention mechanisms (e.g., Performer, ViTaLiTy) from being applied to LLM KV-cache eviction without retraining. By introducing PTAA and aligning normalization criteria, SLAKE enables the reuse of FlashAttention kernels for unevicted tokens, achieving both high approximation accuracy and computational efficiency.
>
> **2. Robustness and Safety Analysis (Table A)**
>
> Addressing the need for safety-critical analysis, we measured the standard deviation and maximum performance degradation across LongBench tasks. SLAKE demonstrates the lowest performance variance and maximum error drop compared to baselines (H2O, PyramidKV, CAKE), proving it provides stable and predictable behavior even in worst-case scenarios.
>
> **3. Automated VABS Hyperparameters (Table B)**
>
> To eliminate concerns about hyperparameter fragility and manual tuning efforts:
> * We verified that performance is insensitive within a broad search range ($\alpha$, $\beta$ $\in$ [$0.1$, $0.7$ ]).
> * Significantly, we introduced an automated regression model that predicts optimal hyperparameters based on model configurations (e.g., embedding dimension, cache budget). This approach yielded SOTA performance on Qwen2.5-32B without any manual search, confirming the method's practicality.
>
>
> **4. Scalability: Large Models and Infinite Contexts (Tables C-1, C-2)**
> * **Long Contexts:** We evaluated SLAKE on InfiniteBench (up to 1M+ tokens). SLAKE achieved the highest average score, demonstrating that approximation errors do not accumulate destructively over extremely long sequences.
> * **Large Models:** New experiments on Llama2-13B and Qwen2.5-32B confirm that SLAKE consistently outperforms state-of-the-art eviction methods as model size increases.
>
> **5. Generalization Capabilities (Table D)**
>
> We extended our evaluation to multilingual tasks (Chinese LongBench) and reaffirmed performance on code generation. SLAKE consistently surpasses existing methods across diverse domains, indicating that our attention approximation remains robust regardless of language or data modality.
>
> We believe these comprehensive evaluations confirm that SLAKE is a statistically robust, scalable, and easily deployable solution for efficient LLM inference. We hope these updates aid in your final assessment.

---

### Author Response · Authors · 2025-12-04
**Final Clarification for the Area Chair (1/2)**

Dear AC,

Thank you for your time and effort in managing this submission.
With full respect for you and all the reviewers, we would like to provide a consolidated summary of our rebuttal and active discussions to assist in your final decision.

---

## **Brief Summary**

- Overall, the reviewers recognized the novelty of SLAKE in unifying linear attention with KV-cache eviction.

- The primary concerns centered on scalability to larger models, hyperparameter sensitivity, and system-level motivation.
We have successfully addressed these through extensive new experiments (scaling up to Qwen2.5-32B, Llama2-13B, and InfiniteBench[1]), the introduction of an automated hyperparameter tuning method, and theoretical clarifications.

- Reviewer **xhuW** raised their score (**2$\rightarrow$4**) following our discussion, and Reviewers **MgG5** and **4zgC** have maintained positive assessments (Score 6).

---

## **Response to Reviewer 1mfi (Score 4)**

### **W1: Novelty & Technical Contribution**
We clarified that SLAKE is not a trivial combination but a fundamental resolution to the structural conflict between Softmax normalization and Taylor linearization. This allows standard FlashAttention kernels to be reused for unevicted tokens, ensuring both speed and accuracy without retraining.

### **W2: Robustness and Safety Analysis (Table A)**
Addressing safety-critical concerns, we measured performance variance across LongBench tasks. SLAKE demonstrates the lowest performance variance and maximum error drop compared to baselines (H2O[2], PyramidKV[3], CAKE[4]), proving predictable behavior in worst-case scenarios.

### **W3 & Q3: Automated VABS Hyperparameters**
To eliminate concerns about fragility:
* **Stability:** We verified performance is insensitive within a broad range ($\alpha, \beta \in [0.1, 0.7]$).
* **Automation:** We introduced an automated regression model that predicts optimal hyperparameters based on model configurations. This achieved SOTA performance on Qwen2.5-32B without manual tuning.

### **Q1 & Q2: Scalability & Generalization**
* **Long Contexts:** SLAKE achieved the highest average score on InfiniteBench[1] (up to 1M+ tokens).
* **Large Models:** New experiments on Llama2-13B and Qwen2.5-32B confirm SLAKE consistently outperforms baselines as model size increases.
* **Generalization:** We validated robustness on multilingual (Chinese) and code generation tasks (Table D)[5].

---

## **Response to Reviewer xhuW (Score 2$\rightarrow$4)**
We appreciate the active engagement from Reviewer xhuW, whose feedback contributed to a score increase from 2 to 4.

### **W1: Motivation & System Perspective**
The reviewer questioned the benefit of linear attention given that standard attention is often memory-bound. We clarified that while full-cache inference is memory-bound, eviction-based inference shifts the bottleneck to accuracy recovery (information loss). SLAKE uses PTAA to mathematically reconstruct evicted tokens with negligible computational overhead (~1.97 MFLOPs), effectively balancing system efficiency with generation quality.

### **W2: Scalability to Larger Models**
Addressing the request for validation on larger models (13B/30B), we provided new experimental results on Llama2-13B and Qwen2.5-32B. SLAKE consistently outperforms state-of-the-art eviction methods on these larger architectures, proving that our method scales seamlessly without structural limitations.

### **W3: Novelty & Practicality**
We emphasized that SLAKE is not a trivial combination of existing ideas (e.g., ViTALiTy[6]). Unlike prior works that require retraining or are limited to vision tasks, SLAKE resolves the structural conflict between Softmax normalization and Taylor linearization. This enables a training-free, drop-in replacement that is practically deployable with our newly introduced automated hyperparameter tuning.

---

## **Response to Reviewer MgG5 (Score 6)**
### **W1 & Q1: Computational Cost (No Hidden $O(N^2)$)**
We theoretically proved that PTAA and VABS operations are strictly linear ($O(N \cdot K)$) as they operate on local tokens or reuse existing attention scores. Empirical data confirms PTAA adds only 137.88 MFLOPs (prefill) and 1.97 MFLOPs (decoding), preserving the speed advantage.

### **W2: Robustness on NeedleBench**
We provided a fair comparison under identical cache budgets (Table D), showing that SLAKE achieves superior retrieval accuracy compared to H2O[2] and CAKE[4].

### **Q3: Speedup Clarification**
We clarified that the 10x speedup refers to token-level decoding latency. Even considering end-to-end latency (including prefill), SLAKE delivers a substantial 5.8x speedup.

---

> ### Author Response · Authors · 2025-12-04
> **Final Clarification for the Area Chair (2/2)**
>
> ## **Response to Reviewer 4zgC (Score 6)**
> ### **W1: Component Isolation (Ablation Study)**
> We conducted additional ablation studies on Mistral-7B-v0.3 (Table A). The results confirm that VABS and PTAA provide independent and additive performance gains, validating their orthogonal contributions.
>
> ### **W2: Taylor Expansion Order**
> We clarified that a first-order Taylor expansion is mathematically necessary to maintain linear complexity. Higher-order approximations would revert the cost to quadratic levels.
>
> ### **W3: Reproducibility & Experimental Details**
> We provided the full set of decoding hyperparameters (Temperature=1, Top-p=None) and task-specific generation lengths as requested, ensuring full reproducibility of our results.
>
> ---
>
> We believe these comprehensive evaluations confirm that SLAKE is a statistically robust, scalable, and easily deployable solution for efficient long-context LLM inference.
>
> Sincerely,
> Authors of Submission #15640
>
> ---
>
> ### **references**
>
> [1] Xinrong Zhang, Yingfa Chen, Shengding Hu, Zihang Xu, Junhao Chen, Moo Hao, Xu Han, Zhen Thai, Shuo Wang, Zhiyuan Liu, et al. ∞ bench: Extending long context evaluation beyond 100k tokens. In Proceedings of the 62nd Annual Meeting of the Association for Computational Linguistics (Volume 1: Long Papers), pp. 15262–15277, 2024b.
>
> [2] Zhenyu Zhang, Ying Sheng, Tianyi Zhou, Tianlong Chen, Lianmin Zheng, Ruisi Cai, Zhao Song, Yuandong Tian, Christopher R´e, Clark Barrett, et al. H2o: Heavy-hitter oracle for efficient generative inference of large language models. Advances in Neural Information Processing Systems, 36:34661–34710, 2023.
>
> [3] Dongjie Yang, XiaoDong Han, Yan Gao, Yao Hu, Shilin Zhang, and Hai Zhao. Pyramidinfer: Pyramid kv cache compression for high-throughput llm inference. arXiv preprint arXiv:2405.12532, 2024.
>
> [4] Ziran Qin, Yuchen Cao, Mingbao Lin, Wen Hu, Shixuan Fan, Ke Cheng, Weiyao Lin, and Jianguo Li. Cake: Cascading and adaptive kv cache eviction with layer preferences. arXiv preprint arXiv:2503.12491, 2025.
>
> [5] Yushi Bai, Xin Lv, Jiajie Zhang, Hongchang Lyu, Jiankai Tang, Zhidian Huang, Zhengxiao Du, Xiao Liu, Aohan Zeng, Lei Hou, et al. Longbench: A bilingual, multitask benchmark for long context understanding. arXiv preprint arXiv:2308.14508, 2023.
>
> [6] Jyotikrishna Dass, Shang Wu, Huihong Shi, Chaojian Li, Zhifan Ye, Zhongfeng Wang, and Yingyan Lin. Vitality: Unifying low-rank and sparse approximation for vision transformer acceleration with a linear taylor attention. In 2023 IEEE International Symposium on High-Performance Computer Architecture (HPCA), pp. 415–428. IEEE, 2023.

---

### Meta-Review · Area_Chair_bNSh · 2026-01-10

**Summary:**

The paper introduces a training-free linear attention method for long-seqence language modeling. The key idea is a bit ad-hoc, with softmax appximated as a taylor expansion and additional KV-Cache eviction. Both ideas are heavily explored in the existing literature. Two reviewers explicitly mentioned the novelty weakness. The other reviewers are concerned with the limited empirical valiation.

I have read the paper and the rebuttal. I think some baselines (say H2O) are not compared in a fair setting, as they are only KV compression methods, while this paper further combines an approximation to softmax and make the attention linearly scaled. I find it less convincing as H2O can also leverage similar approximation strategies.

Overall, I tend to reject this paper due to limited novelty and insufficient empirical valiation. It is ok for novelty to be ad-hoc, if the empirical validation is very comprehensive and really shows the practical usefulness of the proposed method. However, the paper, at its current form, fails to achieve this.

**Reviewer Concerns:**

Two reviewers with weak rejction rating criticize the novelty, as the major contribution is to combining linear attention approximation and KV compression. The other two reviewers with positive rating are concerned about its empirical study. I find concerns from both sides make sense to me, and I think the paper needs another round of evaluation before it can be published.

**Reviewer Scores:**

Two positive rating and two negative rating. Overall boarderline score.

---

### Decision · Program_Chairs · 2026-01-26

Reject